# GENERALIZATION THROUGH VARIANCE: HOW NOISE SHAPES INDUCTIVE BIASES IN DIFFUSION MODELS

**John J. Vastola**
Department of Neurobiology
Harvard Medical School
Boston, MA 02115, USA
`john_vastola@hms.harvard.edu`

## ABSTRACT

How diffusion models generalize beyond their training set is not known, and is somewhat mysterious given two facts: the optimum of the denoising score matching (DSM) objective usually used to train diffusion models is the score function of the training distribution; and the networks usually used to learn the score function are expressive enough to learn this score to high accuracy. We claim that a certain feature of the DSM objective—the fact that its target is not the training distribution's score, but a noisy quantity only equal to it in expectation—strongly impacts whether and to what extent diffusion models generalize. In this paper, we develop a mathematical theory that partly explains this 'generalization through variance' phenomenon. Our theoretical analysis exploits a physics-inspired path integral approach to compute the distributions typically learned by a few paradigmatic under- and overparameterized diffusion models. We find that the distributions diffusion models effectively learn to sample from resemble their training distributions, but with 'gaps' filled in, and that this inductive bias is due to the covariance structure of the noisy target used during training. We also characterize how this inductive bias interacts with feature-related inductive biases.

## 1 INTRODUCTION

Diffusion models (Sohl-Dickstein et al., 2015; Song & Ermon, 2019; Ho et al., 2020; Yang et al., 2023) have proven effective at producing high-quality samples (e.g., images) *like* those from some training distribution, but not overwhelmingly so. This ability to generalize is somewhat surprising for two reasons. First, the optimum of the denoising score matching (DSM) objective usually used to train them is the score function of the training distribution (Vincent, 2011; Song & Ermon, 2019), and sampling using this score only reproduces training examples (see Appendix A). Second, the network architectures usually used for score function approximation are highly expressive. Two near-SOTA models developed by Karras et al. (2022) have $\sim 56$ million (CIFAR-10, trained on 200 million samples) and $\sim 296$ million parameters (ImageNet-64, trained on 2500 million samples), respectively. Sufficiently expressive models can fit even random noise (Zhang et al., 2017).

A body of empirical work bears on the question of when and to what extent diffusion models generalize. Training data is more likely to be memorized when training sets are small (Somepalli et al., 2023a; Stein et al., 2023; Dar et al., 2024; Kadkhodaie et al., 2024), contain duplicates (Somepalli et al., 2023a; Carlini et al., 2023; Somepalli et al., 2023b), or feature low 'complexity' (Somepalli et al., 2023b; Stein et al., 2023). The specific training examples more likely to be memorized are either highly duplicated or outliers (Carlini et al., 2023). Whether generalization happens also strongly depends on model capacity, with Yoon et al. (2023) and Zhang et al. (2024) observing a sharp transition from memorization to generalization as the number of training examples used somewhat outstrips model capacity. However, the relationship between model performance (e.g., FID score) and model size, given a fixed number of training examples, is not monotonic; Karras et al. (2024) observe that their ImageNet models strictly improve (and hence generalize better) as model size increases.

At present, there is arguably no theory that describes when diffusion models generalize and characterizes how the associated inductive biases depend on details like training set structure, the choice

of forward/reverse processes, and model architecture. Most existing theoretical work focuses on orthogonal questions: given a *known ground truth*, can one mathematically guarantee that in some limit (e.g., a large or infinite number of samples from the ground truth distribution) diffusion models recover the ground truth, and bound how score approximation error impacts agreement (Bortoli, 2022; Chen et al., 2023a;c; Han et al., 2024)? The question we are interested in is qualitatively different: given $M \geq 1$ examples from a data distribution $p_{data}$, how do samples from a model trained on those examples differ from them? For example, does the model effectively interpolate training data? If so, when, and what details does this depend on? Concurrent work (Kamb & Ganguli, 2024; Niedoba et al., 2025) addresses these questions at the level of phenomenology, but not mechanism.

In this paper, we argue that six factors substantially impact how diffusion models generalize.

1. **Noisy objective.** The target of the DSM objective is not the score of the training distribution, but a noisy quantity *only equal to it in expectation*. This quantity, which we call the 'proxy score', introduces additional randomness to training, and has extremely high variance at low noise levels (infinite variance, in fact, at zero noise). Intuitively, this makes score function estimates, especially at low noise levels, inaccurate (this is well-known; Karras et al. (2022) remark on this when they discuss their choice of loss weighting). Moreover, this variance is not uniform in state space, but higher in 'boundary regions', e.g., regions of state space close to multiple training examples. This provides a useful inductive bias.

2. **Forward process.** Details of the forward process (e.g., when noise is added, asymmetry in how noise is added along different directions of state space) affect generalization through their influence on the covariance structure of the proxy score.

3. **Nonlinear score-dependence.** The learned distribution depends nonlinearly on the learned score function through the dynamics of the reverse process. This implies that the average learned distribution is *not* the training distribution, even if the score estimator is unbiased.

4. **Model capacity.** Models generalize better when # training samples $\sim$ # model parameters.

5. **Model features.** Feature-related inductive biases interact with, and can enhance, inductive biases due to the covariance structure of the proxy score.

6. **Training set structure.** Nontrivial generalization (e.g., interpolation) is substantially more likely when a large number of training examples are near each other in state space; outliers are less likely to be meaningfully generalized.

Hence, details of training (1, 2), sampling (3), model architecture (4, 5), and the training set (6) all interact to determine the details of generalization. Other aspects, like learning dynamics, also almost certainly play a role, but we mostly neglect them here. The first factor is particularly important, and without it we will see that diffusion models do not generalize well; for this reason, we refer to the phenomenon enabled by (1) and affected by (2-6) as **generalization through variance**.

We support this claim using physics-inspired theory. The Martin-Siggia-Rose (MSR) path integral description of stochastic dynamics (Martin et al., 1973), which has also been exploited to characterize random neural networks (Crisanti & Sompolinsky, 2018) and learning dynamics (Mignacco et al., 2020; Bordelon & Pehlevan, 2022; 2023), plays a pivotal role in our analysis. First, we use the MSR path integral to derive the generic form of 'generalization through variance', and then we discuss in specific, analytically tractable cases of interest (e.g., linear models, lazy infinite-width neural networks) how the details change and the role of each of the aforementioned factors. To keep our theoretical analysis tractable, we focus on unconditional, non-latent models.

## 2 PRELIMINARIES

**Data distribution.** Let $p_{data}(\boldsymbol{x}_0)$ denote a data distribution on $\mathbb{R}^D$. We are especially but not exclusively interested in the case that $p_{data}$ consists of a discrete set of $1 \leq M < \infty$ examples (e.g., images), so that $p_{data}(\boldsymbol{x}_0) = \sum_{m=1}^{M} \delta(\boldsymbol{x}_0 - \boldsymbol{\mu}_m)/M$, where $\delta$ is the Dirac delta function.

**Forward/reverse diffusion.** Training a diffusion model involves learning to convert samples from a normal distribution $p_{noise}(\boldsymbol{x}_T) = \mathcal{N}(\boldsymbol{x}_T; \boldsymbol{0}, \boldsymbol{S}_T)$ to samples from $p_{data}(\boldsymbol{x}_0)$ via processes

$$\dot{\boldsymbol{x}}_t = -\beta_t \boldsymbol{x}_t + \boldsymbol{G}_t \boldsymbol{\eta}_t \qquad t = 0 \rightarrow t = T \qquad \text{forward process, } p_{data} \text{ to } p_{noise} \qquad (1)$$

$$\dot{\boldsymbol{x}}_t = -\beta_t \boldsymbol{x}_t - \boldsymbol{D}_t \boldsymbol{s}(\boldsymbol{x}_t, t) \qquad t = T \rightarrow t = \epsilon \qquad \text{reverse process, } p_{noise} \text{ to } p_{data} \qquad (2)$$

Table 1: Popular forward processes in our parameterization. For these, $\boldsymbol{G}_t := g_t \boldsymbol{I}_D$ and $\boldsymbol{S}_t = \sigma_t^2 \boldsymbol{I}_D$.

| | $\beta_t$ | $g_t$ | $\alpha_t$ | $\sigma_t$ | end time |
|---|---|---|---|---|---|
| VP-SDE | $\beta_{min} + \beta_d t$ | $\sqrt{2\beta_t}$ | $e^{-\int_0^t \beta_{t'}\, dt'}$ | $\sqrt{1 - e^{-2\int_0^t \beta_{t'}\, dt'}}$ | 1 |
| EDM | 0 | $\sqrt{2t}$ | 1 | $t$ | $T$ |

where $\boldsymbol{\eta}_t \in \mathbb{R}^K$ is Gaussian white noise, $\boldsymbol{G}_t \in \mathbb{R}^{D \times K}$ is a nonnegative matrix that controls the noise amplitude, $\boldsymbol{D}_t := \boldsymbol{G}_t \boldsymbol{G}_t^T / 2$ is the corresponding diffusion tensor, $\beta_t \geq 0$ controls decay to the origin, $\epsilon > 0$ is a time cutoff that helps ensure numerical stability, and $\boldsymbol{s}(\boldsymbol{x}, t) := \nabla_{\boldsymbol{x}} \log p(\boldsymbol{x}|t)$ is the score function. We allow $\boldsymbol{G}_t$ to be a matrix so we can study how asymmetries affect generalization later. The forward process' marginals are $p(\boldsymbol{x}|t) := \int p(\boldsymbol{x}|\boldsymbol{x}_0, t) p_{data}(\boldsymbol{x}_0)\, d\boldsymbol{x}_0$. The transition probabilities are $p(\boldsymbol{x}|\boldsymbol{x}_0, t) = \mathcal{N}(\boldsymbol{x}; \alpha_t \boldsymbol{x}_0, \boldsymbol{S}_t)$, where $\alpha_t := e^{-\int_0^t \beta_{t'}\, dt'}$ and $\boldsymbol{S}_t := \int_0^t 2\boldsymbol{D}_{t'} \alpha_{t'}^2\, dt'$.

The forward process assumed here is fairly general, and includes popular choices like the VP-SDE (Song et al., 2021) and EDM formulation (Karras et al., 2022) (Table 1). This choice of reverse process is called the probability flow ODE (PF-ODE), and has been shown to have both practical (Song et al., 2021) and theoretical (Chen et al., 2023b) advantages. Since $\boldsymbol{s}(\boldsymbol{x}, t)$ is required to run the reverse process but is a priori unknown, "training" a model means approximating $\boldsymbol{s}(\boldsymbol{x}, t)$.

**Denoising score matching.** Given $P \gg 1$ independent samples from $p(\boldsymbol{x}, \boldsymbol{x}_0, t)$ (note: $P$ is different than $M$, the number of points in discrete $p_{data}$), one could use a mean-squared-error objective

$$J_0(\boldsymbol{\theta}) := \mathbb{E}_{t, \boldsymbol{x}} \left\{ \frac{\lambda_t}{2} \|\hat{\boldsymbol{s}}_{\boldsymbol{\theta}}(\boldsymbol{x}, t) - \boldsymbol{s}(\boldsymbol{x}, t)\|_2^2 \right\} = \int \frac{\lambda_t}{2} \|\hat{\boldsymbol{s}}_{\boldsymbol{\theta}}(\boldsymbol{x}, t) - \boldsymbol{s}(\boldsymbol{x}, t)\|_2^2\, p(\boldsymbol{x}|t)p(t)\, d\boldsymbol{x}dt \quad (3)$$

to learn a parameterized score estimator $\hat{\boldsymbol{s}}_{\boldsymbol{\theta}}(\boldsymbol{x}, t)$. Here, $\lambda_t > 0$ is a positive weighting function and $p(t)$ is a time-sampling distribution. The DSM objective (Vincent, 2011; Song & Ermon, 2019)

$$J_1(\boldsymbol{\theta}) := \mathbb{E}_{t, \boldsymbol{x}_0, \boldsymbol{x}} \left\{ \frac{\lambda_t}{2} \|\hat{\boldsymbol{s}}_{\boldsymbol{\theta}}(\boldsymbol{x}, t) - \tilde{\boldsymbol{s}}(\boldsymbol{x}, t; \boldsymbol{x}_0)\|_2^2 \right\} = \int \frac{\lambda_t}{2} \|\hat{\boldsymbol{s}}_{\boldsymbol{\theta}} - \tilde{\boldsymbol{s}}\|_2^2\, p(\boldsymbol{x}, \boldsymbol{x}_0, t)\, d\boldsymbol{x}d\boldsymbol{x}_0 dt \quad (4)$$

where $p(\boldsymbol{x}, \boldsymbol{x}_0, t) := p(\boldsymbol{x}|\boldsymbol{x}_0, t) p_{data}(\boldsymbol{x}_0) p(t)$, is usually used instead. While the folklore justifying this choice is that the score function is not known, this is not true; both $J_0$ and $J_1$ are optimized when $\hat{\boldsymbol{s}}_{\boldsymbol{\theta}}$ equals the score of the training distribution (see Appendix A), which is known.

We will argue that the real difference between $J_0$ and $J_1$ is that $J_1$ generalizes better, and that this is in part because the **proxy score** $\tilde{\boldsymbol{s}}(\boldsymbol{x}, t; \boldsymbol{x}_0) := \nabla_{\boldsymbol{x}} \log p(\boldsymbol{x}|\boldsymbol{x}_0, t) = \boldsymbol{S}_t^{-1}(\alpha_t \boldsymbol{x}_0 - \boldsymbol{x})$ is used as the target instead of the true score. It is a 'noisy' version of the true score (see Appendix B), since

$$\mathbb{E}_{\boldsymbol{x}_0|\boldsymbol{x}, t}[\tilde{\boldsymbol{s}}(\boldsymbol{x}, t; \boldsymbol{x}_0)] = \boldsymbol{s}(\boldsymbol{x}, t) \qquad C_{ij}(\mathbf{x}, t) := \mathrm{Cov}_{\boldsymbol{x}_0|\boldsymbol{x}, t}[\tilde{s}_i, \tilde{s}_j] = S_{t, ij}^{-1} + \partial_{ij}^2 \log p(\boldsymbol{x}|t) . \quad (5)$$

Although the proxy score is equal to the score of the training distribution in expectation, neural networks trained on $J_1$ empirically learn a different distribution and generalize better. We claim that this fact is closely related to the *covariance structure* of the proxy score. Two relevant observations about its form are as follows. First, it is large at small times, since $\boldsymbol{S}_t \to \boldsymbol{0}$ as $t \to 0$. Second, it is large where the log-likelihood $\log p(\boldsymbol{x}|t)$ has substantial curvature. In the typical case, where $p_{data}$ consists of a discrete set of $M$ examples, regions of high curvature correspond to the location of training examples and the boundaries between them (Fig. 1; see Appendix C for more discussion).

**Generalization and inductive biases.** In a typical supervised learning setting, one trains a model on one set of data and tests it on another, and defines 'generalization error' as performance on the held-out data. Here, we are interested in a different type of problem: *given* a model trained on samples from $p(\boldsymbol{x}, \boldsymbol{x}_0, t)$, *to what extent* does the learned distribution differ from $p_{data}$, and what are the associated inductive biases? Of particular interest is whether models do three things: (i) interpolation (filling in gaps in the training data), (ii) extrapolation (extending patterns in the training data), and (iii) feature blending (generating samples which include both feature $X$ and feature $Y$ even when training examples only involve one of the two features).

In our setting, a subtle but important point is that there is generally no ground truth. For example, the smooth distribution that CIFAR-10 or MNIST images are drawn from does not exist, except in a 'Platonic' sense; we are interested in the extent to which diffusion models learn a distribution plausibly *like* a smoothed version of the training distribution.

Figure 1: Visualization of proxy score variance $(\mathrm{tr}(\boldsymbol{C})/[\mathrm{tr}(\boldsymbol{C}) + \|\boldsymbol{s}\|_2^2])$ for four example 2D data distributions. Each example data distribution is supported on a small number of point masses (red dots). As $t$ changes (left: small $t$, right: large $t$), boundary regions at different scales are emphasized.

## 3 APPROACH: COMPUTING TYPICAL LEARNED DISTRIBUTIONS

The distribution $q(\boldsymbol{x}_0|\boldsymbol{\theta})$ learned by a diffusion model depends on the learned score $\hat{\boldsymbol{s}}_{\boldsymbol{\theta}}$ nonlinearly through PF-ODE dynamics; importantly, we are less interested in how well the score is estimated, and more interested in how estimation errors impact $q$. The learned score can be viewed as a random variable, since it depends on the $P$ samples $\boldsymbol{x}^{(i)}, \boldsymbol{x}_0^{(i)}, t^{(i)} \sim p(\boldsymbol{x}, \boldsymbol{x}_0, t)$ used during training. In order to theoretically understand how diffusion models generalize, we aim to obtain an analytic expression for the 'typical' learned distribution by averaging $q$ over sample realizations.

How do we do the required averaging? One of our major contributions is to introduce a theoretical approach for averaging $q(\boldsymbol{x}_0)$ over variation due to $\hat{\boldsymbol{s}}$. Below, we describe our approach.

**Writing PF-ODE dynamics in terms of a path integral.** How does one average over the result of an ODE given that, in the case of PF-ODE dynamics, there is generally no closed-form expression for the result? To address this issue, we use a novel **stochastic path integral** representation of PF-ODE dynamics that makes the required average easy to do. If $q(\boldsymbol{x}_0|\boldsymbol{x}_T; \boldsymbol{\theta})$ denotes the distribution of PF-ODE outputs given a score estimator $\hat{\boldsymbol{s}}_{\boldsymbol{\theta}}(\boldsymbol{x}, t)$ and a fixed noise seed $\boldsymbol{x}_T$,

$$q(\boldsymbol{x}_0|\boldsymbol{x}_T; \boldsymbol{\theta}) = \int \mathcal{D}[\boldsymbol{p}_t]\mathcal{D}[\boldsymbol{x}_t] \, \exp\left\{ \int_\epsilon^T i\boldsymbol{p}_t \cdot [\dot{\boldsymbol{x}}_t + \beta_t\boldsymbol{x}_t + \boldsymbol{D}_t\hat{\boldsymbol{s}}_{\boldsymbol{\theta}}(\boldsymbol{x}_t, t)] \, dt \right\} \tag{6}$$

where the integral is over all possible paths from $\boldsymbol{x}_T$ to $\boldsymbol{x}_0$. (To avoid technical issues, we assume a particular time discretization in all calculations. See Appendix D.) This type of path integral is a time-reversed version of the Martin-Siggia-Rose (MSR) path integral (Martin et al., 1973).

**Averaging over possible sample realizations.** Because the argument of the exponential depends linearly on the score, the required ensemble average is now easy to do. Using $[\cdots]$ to denote it,

$$[q(\boldsymbol{x}_0|\boldsymbol{x}_T)] = \int \mathcal{D}[\boldsymbol{p}_t]\mathcal{D}[\boldsymbol{x}_t]\exp\left\{ M_1 - \frac{1}{2}M_2 + \cdots \right\} \tag{7}$$

$$M_1 := \int_\epsilon^T i\boldsymbol{p}_t \cdot [\dot{\boldsymbol{x}}_t + \beta_t\boldsymbol{x}_t + \boldsymbol{D}_t\boldsymbol{s}_{avg}(\boldsymbol{x}_t, t)] \, dt \qquad M_2 := \int_\epsilon^T \int_\epsilon^T \boldsymbol{p}_t^T \boldsymbol{V}(\boldsymbol{x}_t, t; \boldsymbol{x}_{t'}, t')\boldsymbol{p}_{t'} \, dt dt'$$

where $\boldsymbol{s}_{avg}(\boldsymbol{x}_t, t) := [\hat{\boldsymbol{s}}_{\boldsymbol{\theta}}(\boldsymbol{x}_t, t)]$ is the ensemble's average score estimator, and $\boldsymbol{V}(\boldsymbol{x}_t, t; \boldsymbol{x}_{t'}, t') := \boldsymbol{D}_t\mathrm{Cov}_{\boldsymbol{\theta}}[\hat{\boldsymbol{s}}(\boldsymbol{x}_t, t), \hat{\boldsymbol{s}}(\boldsymbol{x}_{t'}, t')]\boldsymbol{D}_{t'}$ measures ensemble variance. Assuming higher-order terms can be neglected—and hence that the estimator distribution is approximately Gaussian—one can show (see Appendix D) that sampling from $[q(\boldsymbol{x}_0|\boldsymbol{x}_T)]$ is equivalent to integrating an (Ito-interpreted) SDE:

**Proposition 3.1** (Effective SDE description of typical learned distribution)**.** *Sampling from* $[q(\boldsymbol{x}_0|\boldsymbol{x}_T)]$ *is equivalent to integrating the (Ito-interpreted) SDE*

$$\dot{\boldsymbol{x}}_t = -\beta_t\boldsymbol{x}_t - \boldsymbol{D}_t\boldsymbol{s}_{avg}(\boldsymbol{x}_t, t) + \boldsymbol{\xi}(\boldsymbol{x}_t, t) \qquad\qquad t = T \to t = \epsilon \tag{8}$$

*with initial condition* $\boldsymbol{x}_T$*, where* $\boldsymbol{s}_{avg}(\boldsymbol{x}_t, t) := [\hat{\boldsymbol{s}}_{\boldsymbol{\theta}}(\boldsymbol{x}_t, t)]$ *and where the noise term* $\boldsymbol{\xi}(\boldsymbol{x}_t, t)$ *has mean zero and autocorrelation* $\boldsymbol{V}(\boldsymbol{x}_t, t; \boldsymbol{x}_{t'}, t') := \boldsymbol{D}_t Cov_{\boldsymbol{\theta}}[\hat{\boldsymbol{s}}(\boldsymbol{x}_t, t), \hat{\boldsymbol{s}}(\boldsymbol{x}_{t'}, t')]\boldsymbol{D}_{t'}$.

If $\hat{\boldsymbol{s}}$ is unbiased and $M$ is finite, then the noise term is solely responsible for the difference between true PF-ODE dynamics (which reproduces training examples) and a model's 'typical' sampling

dynamics—i.e., generalization occurs if and only if $V \neq 0$. This makes characterizing $V$, which we call the *V-kernel* since it reflects ensemble variance, crucially important for understanding how diffusion models generalize. Our remaining theoretical work is to complete two tasks: first, to compute $s_{avg}$ and $V$ for a few paradigmatic and theoretically tractable architectures; and second, to study how their precise forms affect $[q(x_0)]$.

## 4 DIFFUSION MODELS THAT MEMORIZE TRAINING DATA STILL GENERALIZE

It is instructive to first consider an extreme case: do diffusion models generalize in the complete *absence* of any model-related inductive biases? Perhaps surprisingly, the answer is yes. In this section, we make this point using a toy model in which training and sampling are interleaved.

Suppose the PF-ODE is integrated backward in time from an initial point $x_T$ until $t = \epsilon$ using first-order Euler updates of size $\Delta t$. At each time step, suppose one samples $x_{0t} \sim p(x_0|x_t, t) = \frac{p(x_t|x_0, t)p_{data}(x_0)}{p(x_t|t)}$, constructs the 'naive' score estimator $\hat{s}(x_t, t) := s(x_t, t) + \sqrt{\frac{\kappa}{\Delta t}}[\tilde{s}(x_t, t; x_{0t}) - s(x_t, t)]$, and uses this estimator as the score for that update. Assume this process continues, with new samples drawn at each time step. Despite this approach using the proxy score directly (so that training data is 'memorized'), one obtains a nontrivial V-kernel, and hence generalization:

**Proposition 4.1** (Naive score estimator generalizes). *Consider the result of integrating the PF-ODE (Eq. 2) from $t = T$ to $t = \epsilon$ using first-order Euler updates of the form*

$$x_{t-1} = x_t + \Delta t \left\{ \beta_t x_t + D_t \left( s(x_t, t) + \sqrt{\frac{\kappa}{\Delta t}}[\tilde{s}(x_t, t; x_{0t}) - s(x_t, t)] \right) \right\} , \quad x_{0t} \sim p(x_0|x_t, t) .$$

*Then $[q(x_0|x_T)]$ is described by an effective SDE (Eq. 8) with $s_{avg} = s$ and V-kernel*

$$V(x_t, t; x_{t'}, t') := \kappa D_t C(x, t) D_t \, \delta(t - t') . \tag{9}$$

See Appendix E for details. Notably, the effective SDE is noisier when the covariance $C$ of the proxy score is high, e.g., in boundary regions between training examples. Next, we will see that this is also true for less trivial models, but that the proxy score's covariance interacts with feature-related biases in order to determine the SDE's overall noise term.

## 5 FEATURE-RELATED INDUCTIVE BIASES MODULATE GENERALIZATION

Model architecture is known to produce certain inductive biases, with spectral bias being a well-known example (Rahaman et al., 2019; Bordelon et al., 2020; Canatar et al., 2021). How do model-feature-related inductive biases affect the V-kernel? We answer this question below in two interesting but tractable cases: linear models, and (lazy regime) infinite-width neural networks.

### 5.1 THE V-KERNEL OF EXPRESSIVE LINEAR MODELS

In what follows, we may write $z := (x, t)$ to ease notation. Consider a linear score estimator

$$\hat{s}_\theta(x, t) = w_0 + W \phi(x, t) , \tag{10}$$

where the $F$ feature maps $\phi := (\phi_1, ..., \phi_F)^T$ are linearly independent, smooth functions from $\mathbb{R}^D \times (0, T]$ to $\mathbb{R}$ that are square-integrable with respect to the measure $\lambda_t p(x, t)$. The parameters to be estimated are $\theta := \{w_0, W\}$, with $w_0 \in \mathbb{R}^D$ and $W \in \mathbb{R}^{D \times F}$. Note that this estimator is linear in its features, but not necessarily in $x$ or $t$. The weights that optimize Eq. 4 are (see Appendix F)

$$W^* = -J^T \Sigma_\phi^{-1} \qquad w_0^* = J^T \Sigma_\phi^{-1} \langle \phi \rangle + \langle \tilde{s} \rangle \tag{11}$$

where we define $\langle \cdots \rangle := \mathbb{E}_{x, x_0, t}[\lambda_t \cdots] / \mathbb{E}_t[\lambda_t]$ and matrices

$$J := -\langle [\phi(x, t) - \langle \phi \rangle] [\tilde{s}(x, t; x_0) - \langle \tilde{s} \rangle]^T \rangle \qquad \Sigma_\phi := \langle [\phi(x, t) - \langle \phi \rangle] [\phi(x, t) - \langle \phi \rangle]^T \rangle .$$

When averaged over $x_0$ sample realizations, the estimator $\hat{s}_*(x, t) = w_0^* + W^* \phi(x, t)$ is unbiased as long as the set of feature maps is sufficiently expressive. Interestingly, this is true regardless of the $x$ or $t$ samples used, provided $F \leq P$. The following result characterizes $[q(x_0)]$ for linear models:

**Proposition 5.1** (Expressive linear models asymptotically generalize). *Suppose the parameters of an expressive linear score estimator (Eq. 10) with $F$ features are perfectly optimized according to the DSM objective (Eq. 4) using $P$ independent samples from $p(\boldsymbol{x}, \boldsymbol{x}_0, t)$. Define the feature kernel*

$$k(\boldsymbol{z}; \boldsymbol{z}') := \frac{1}{\sqrt{F}} [\boldsymbol{\phi}(\boldsymbol{z}) - \langle \boldsymbol{\phi} \rangle]^T \boldsymbol{\Sigma}_\phi^{-1} [\boldsymbol{\phi}(\boldsymbol{z}') - \langle \boldsymbol{\phi} \rangle] \,. \tag{12}$$

*Let $\kappa := F/P$. Provided that the limit exists and is finite, in the $P \to \infty$ limit, we have*

$$\boldsymbol{V}(\boldsymbol{z}; \boldsymbol{z}') = \lim_{P \to \infty} \kappa \, \boldsymbol{D}_t \, \mathbb{E}_{\boldsymbol{z}''} \left\{ \frac{\lambda_{t''}^2}{\mathbb{E}_t[\lambda_t]^2} k(\boldsymbol{z}; \boldsymbol{z}'') \boldsymbol{C}(\boldsymbol{z}'') k(\boldsymbol{z}''; \boldsymbol{z}') \right\} \boldsymbol{D}_{t'} \,. \tag{13}$$

Note that if the number of features $F$ does not scale with $P$, $\boldsymbol{V} \equiv \boldsymbol{0}$. See Appendix F for the details of our argument. The V-kernel for linear models differs from the naive score's V-kernel (Eq. 9) via the presence of feature-related factors—in particular, the effective SDE is noisier *where features take atypical values*. One expects that these factors can either enhance or compete with noise due to the covariance structure (e.g., noise is higher if features take atypical values in boundary regions).

## 5.2 THE V-KERNEL OF LAZY INFINITE-WIDTH NEURAL NETWORKS

Neural networks in the neural tangent kernel (NTK) regime (Jacot et al., 2018; Bietti & Mairal, 2019) provide another interesting but tractable model. Such networks exhibit 'lazy' learning (Chizat et al., 2019) in the sense that weights do not move much from their initial values. Moreover, it is known that they interpolate training data in the absence of parameter regularization or early stopping (Bordelon et al., 2020). If they precisely interpolated their samples, we would expect to recover a V-kernel like the one we computed in Sec. 4; more generally, we expect something similar modified by the spectral inductive biases associated with the architecture (Canatar et al., 2021).

For simplicity, we consider fully-connected networks whose hidden layers all have width $N$, which is taken to infinity together with $P$ (see Appendix G for details). The associated NTK has a Mercer decomposition with respect to the measure $\lambda_t p(\boldsymbol{x}, t)/\mathbb{E}[\lambda_t]$, so $K$ can be written in terms of $F$ orthonormal features $\{\phi_i\}$:

$$K(\boldsymbol{x}, t; \boldsymbol{x}', t') = \sum_k \lambda_k \phi_k(\boldsymbol{x}, t) \phi_k(\boldsymbol{x}', t') \qquad \int \frac{\lambda_t}{\mathbb{E}[\lambda_t]} \phi_k(\boldsymbol{x}, t) \phi_\ell(\boldsymbol{x}, t) \, p(\boldsymbol{x}, t) \, d\boldsymbol{x} dt = \delta_{k\ell} \,. \tag{14}$$

We assume training involves full-batch gradient descent on $P$ samples from $p(\boldsymbol{x}, \boldsymbol{x}_0, t)$, so that the learned score function after training for an amount of 'time' $\tau$ has the closed-form solution

$$\hat{\boldsymbol{s}}(\boldsymbol{z}) = \hat{\boldsymbol{s}}_0(\boldsymbol{z}) + [\tilde{\boldsymbol{S}} - \hat{\boldsymbol{S}}_0]^T (\boldsymbol{I} - e^{-\boldsymbol{\Lambda}_T \boldsymbol{K} \tau/P}) \boldsymbol{K}^{-1} \boldsymbol{k}(\boldsymbol{z})$$

where $\hat{\boldsymbol{s}}_0$ is the network's initial output, $\tilde{\boldsymbol{S}} \in \mathbb{R}^{D \times P}$ contains proxy score samples, $\hat{\boldsymbol{S}}_0 \in \mathbb{R}^{D \times P}$ contains the network's initial outputs given the samples, $\boldsymbol{K} \in \mathbb{R}^{P \times P}$ is the kernel Gram matrix, $\boldsymbol{\Lambda}_T \in \mathbb{R}^{P \times P}$ is a diagonal matrix containing the weighting function $\lambda_t/\mathbb{E}[\lambda_t]$ evaluated on samples, and $\boldsymbol{k}(\boldsymbol{x}, t)$ is an input-dependent vector whose $i$-th component is $K(\boldsymbol{x}^{(i)}, t^{(i)}; \boldsymbol{x}, t)$. We have:

**Proposition 5.2** (Lazy neural networks asymptotically generalize). *Suppose the parameters of a fully-connected, infinite-width neural network characterized by a rank $F$ NTK are optimized according to the DSM objective (Eq. 4) using $P$ independent samples from $p(\boldsymbol{x}, \boldsymbol{x}_0, t)$ via full-batch gradient descent for a training 'time' $\tau$. Define the feature kernel*

$$k(\boldsymbol{z}; \boldsymbol{z}') := \frac{1}{\sqrt{F}} \boldsymbol{\phi}(\boldsymbol{z})^T (\boldsymbol{I}_F - e^{-\boldsymbol{\Lambda}\tau}) \boldsymbol{\phi}(\boldsymbol{z}') \,. \tag{15}$$

*Let $\kappa := F/P$. Provided that the limit exists and is finite, in the $P \to \infty$ limit, we have*

$$\boldsymbol{V}(\boldsymbol{z}; \boldsymbol{z}') = \lim_{P \to \infty} \kappa \boldsymbol{D}_t \, \mathbb{E}_{\boldsymbol{z}''} \left\{ \frac{\lambda_{t''}^2}{\mathbb{E}[\lambda_t]^2} k(\boldsymbol{z}; \boldsymbol{z}'') \boldsymbol{C}(\boldsymbol{z}'') k(\boldsymbol{z}''; \boldsymbol{z}') \right\} \boldsymbol{D}_{t'} \,. \tag{16}$$

*In the infinite training time limit, we recover the Sec. 4 result with a prefactor $\kappa(\Delta \boldsymbol{z}) = const.$:*

$$\boldsymbol{V}(\boldsymbol{z}; \boldsymbol{z}') = \kappa(\Delta \boldsymbol{z}) \boldsymbol{D}_t \boldsymbol{C}(\boldsymbol{z}) \boldsymbol{D}_t \, \delta(\boldsymbol{z} - \boldsymbol{z}') \,. \tag{17}$$

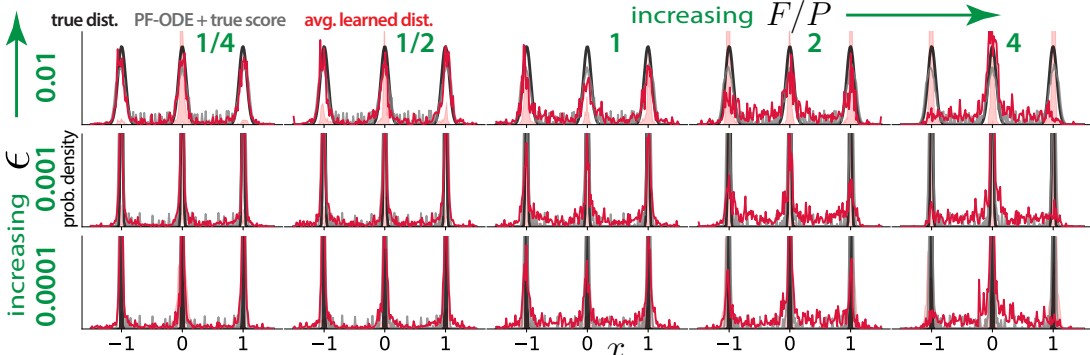

Figure 2: Average learned distribution ($N = 100$) for a linear model with Gaussian features trained on different sample draws from a 1D data distribution $\{-1, 0, 1\}$. Red: average learned distribution; black: true distribution; gray: PF-ODE approximation of true distribution. Different values of the time cutoff $\epsilon$ and ratio $F/P$ are shown. Note that there is more generalization as both become larger.

See Appendix G for the full details of our argument. Interestingly, although the network is not assumed to be in the feature-learning regime, this result interpolates between our pure memorization (Prop. 4.1) and linear model (Prop. 5.1) results as we change the value of the training time $\tau$. The feature-related inductive biases that appear are precisely the well-known spectral biases.

See Appendices H and I for discussion of how to obtain analogous results for diffusion models with slightly different training objectives, like those for which a 'denoiser' rather than a score approximator is learned (see, e.g., Karras et al. (2022)).

## 6 GENERALIZATION THROUGH VARIANCE: CONSEQUENCES AND EXAMPLES

In this section, we briefly discuss salient consequences of generalization through variance.

**Benign properties of generalization through variance.** In what sense might generalization through variance provide a 'reasonable' inductive bias? Its key driver is the proxy score covariance, which is large primarily in boundary regions between training examples (Appendix C), and this fact greatly constrains the way this type of generalization can occur. A data set with one data point (so that $M = 1$) is not generalized, since the proxy score covariance is trivially zero. If the data distribution is primarily supported on some low-dimensional 'data manifold', the proxy score covariance tends to be nontrivial only along that manifold, and hence generalization through variance preserves the dimensionality of the data manifold.

Very far from training examples, the proxy score covariance is approximately zero, so there is no generalization through variance. Finally, the effective PF-ODE both follows deterministic PF-ODE dynamics *on average*, since the V-kernel-related noise term has mean zero, and is also *most likely* to follow deterministic PF-ODE dynamics, since the probability of paths that deviate from it can be shown to be somewhat lower. This means that, although effective PF-ODE dynamics differ from the deterministic PF-ODE's dynamics, they do not *substantially* differ, meaning that regions near training data will still tend to be sampled most. See Appendix J for more details and discussion.

**Memorization and the V-kernel in the small noise limit.** Our characterization of the typical learned distribution $[q]$ in terms of a stochastic process is somewhat unsatisfying, in part because it remains unclear how the V-kernel affects the way $[q]$ generalizes $p_{data}$. One can make some progress on the issue by making a small noise approximation, which is valid (for example) when models are somewhat underparameterized, so that $\kappa = F/P$ is somewhat less than 1. When the effective PF-ODE's noise term is sufficiently small, one can invoke a semiclassical approximation of the relevant path integral. We find (Appendix K) that, at least in this limit,

$$[q(\boldsymbol{x}_0)] \approx p(\boldsymbol{x}_0|\epsilon) \frac{1}{\sqrt{\det\left(\frac{1}{\kappa} \frac{\partial^2 \mathcal{S}_{cl}(\boldsymbol{x}_0, \boldsymbol{x}_T^*(\boldsymbol{x}_0))}{\partial \boldsymbol{x}_T \partial \boldsymbol{x}_T}\right)}} \quad (18)$$

where $\mathcal{S}_{cl}$ quantifies the (negative log-) likelihood of the most likely path that goes from a noise seed $\boldsymbol{x}_T$ to a sample $\boldsymbol{x}_0$. In words: $[q]$ equals the ($\epsilon$-noise-corrupted) data distribution, times a curvature factor that quantifies the likelihood of small deviations from deterministic PF-ODE dynamics. It is through the V-kernel's influence on this curvature term that it affects generalization (although unfortunately it appears difficult to be more explicit about how it does so, at least analytically).

**Gap-filling inductive bias.** Given that the V-kernel is especially sensitive to the 'gaps' between training examples, one expects that generalization through variance works by effectively filling in these gaps. This appears to be often, but not always, true. First, in its naive form (see, e.g., Sec. 4) generalization through variance can actually *reduce* the probability associated with boundary regions, since the additional noise in those regions makes the dynamics spend less time in them. If there is nontrivial temporal generalization, for example via time-dependent features $\phi$, the V-kernel may have a nontrivial temporal autocorrelation structure; we speculate that these autocorrelations may be a key mechanism that allows the dynamics to spend more time in boundary regions.

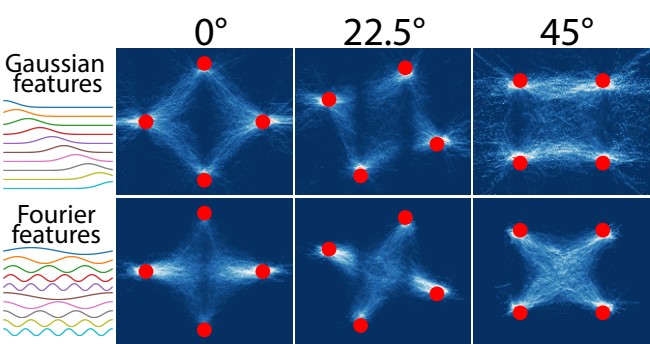

Figure 3: Generalization of a 2D data distribution depends on features used and data orientation. Heatmaps of samples from $N = 100$ linear models are shown in different conditions, with training data (red dots) overlaid. Notice that which gaps are 'filled in', e.g., whether a square shape or cross shape is made, depends on both factors.

Second, the details of generalization are strongly modulated by two numbers: the time cutoff $\epsilon$, and the ratio $F/P$ that determines the extent to which a model is over- or underparameterized. Fig. 2 depicts an illustrative one-dimensional example where there are three training examples $\{-1, 0, 1\}$, and where the model is linear (see Prop. 5.1) with Gaussian features centered at different values of $x$ and $t$, each with the same width. The average learned distribution (red) tends to differ from both the true distribution (black) and its PF-ODE approximation (gray) in the size of peaks near training data, and in the regions between training data. These differences are larger when $F/P$ and $\epsilon$ are larger. Taking both large produces the largest difference, but not obviously the 'best' generalization of training data.

**Feature-noise alignment affects generalization.** Different feature sets interact with the structure of the proxy score covariance differently, and hence produce different kinds of generalization. Fig. 3 shows how the same 2D data distribution (four examples, which together determine the vertices of a square) is generalized differently depending on its orientation, and depending on which linear model feature set (here, either Gaussian or Fourier features) is used.

## 7 DISCUSSION

We used a novel path-integral approach to quantitatively characterize the 'typical' distribution learned by diffusion models, and find that generalization is influenced by a combination of factors related to training (the DSM objective and forward process; Sec. 2 and 4), sampling (the learned distribution depends nonlinearly on score estimates; Sec. 3), model architecture (Sec. 5), and the data distribution. Below, we use our theory to comment on various previous observations.

**DSM produces noisy estimators, but stable distributions.** Various forms of score 'mislearning' are well-known. At small times, scores are hard to learn due to the noisiness of the proxy score target, leading authors like Karras et al. (2022) to suggest a $p(t)$ that emphasizes intermediate noise scales. Chao et al. (2022) discuss how score estimation errors affect conditional scores. Xu et al. (2023) explicitly study the variance-near-mode-boundaries issue we discussed, and propose a strategy for mitigating it. On the other hand, it is well-known that despite noisy score estimates, diffusion models generally produce smooth output distributions (see, e.g., Luzi et al. (2024)). Moreover, two diffusion models trained on non-overlapping subsets of a data set are often highly similar (Kadkhodaie et al., 2024). These facts are due to noisy score estimates contributing to sample generation through the PF-ODE, which effectively 'averages' over estimator noise. Our theory is consistent with these

observations: even the interleaved training-sampling procedure discussed in Sec. 4 produces a well-behaved, smooth distribution.

**DSM produces a boundary-smearing inductive bias.** This has been previously pointed out by authors like Xu et al. (2023). Where we differ from previous authors is in considering this issue a potential strength. Integrating the PF-ODE using the true score reproduces training examples, so it is in some sense beneficial to 'mislearn' the score. This particular kind of mislearning is useful for several ways of generalizing point clouds, including interpolation, extrapolation, and feature blending. Moreover, producing this inductive bias is an interesting way diffusion models differ from something like kernel density estimation: boundary regions *across different noise scales* are smeared out, with different scales linked via PF-ODE dynamics, which may provide better generalization than convolving the training distribution with any single kernel.

**Architecture-related inductive biases play a role.** As we showed in Sec. 5, feature/architecture-related inductive biases interact with DSM's boundary-smearing bias in order to determine how diffusion models generalize. This appears to be consistent, for example, with the Kadkhodaie et al. (2024) finding that diffusion models effectively exhibit 'adaptive geometric harmonic priors'; their finding is specifically in the context of score estimation using a convolutional neural network (CNN) architecture. It is plausible that this choice encourages a harmonic inductive bias, since CNNs more generally exhibit inductive biases related to translation equivariance (Cohen & Welling, 2016).

**Generalization through variance harmful and helpful.** It is important to note that this kind of generalization is not always helpful. A trivial example is that unconditional models trained on MNIST digit images tend to learn to produce non-digits as output in the absence of label information (see, e.g., Bortoli et al. (2021)). More generally, blending modes may or may not be desirable, since it can produce (e.g.) images very qualitatively different from those of the training distribution.

**Other forms of generalization are possible.** Factors we did not study, like learning dynamics, most likely also partly determine how diffusion models generalize. For example, the use of stochastic gradient descent introduces additional randomness that disfavors converging on sharp local optima (Smith & Le, 2018; Smith et al., 2020). It would be interesting to utilize recent theoretical tools (Bordelon & Pehlevan, 2023) to characterize how learning dynamics impacts generalization, especially in the rich (Geiger et al., 2020; Woodworth et al., 2020) rather than lazy learning regime.

**Comment on memorization.** Determining whether diffusion models memorize data (Somepalli et al., 2023a; Carlini et al., 2023), and if so how to address the issue (Vyas et al., 2023), has become a significant technical and societal issue. Our theory suggests that since generalization through variance happens primarily in boundary regions, diffusion models are unlikely to substantially generalize outliers. Since conditional models involve distributions of much higher effective dimension, one may expect that more training examples are 'outlier-like', and hence memorization should happen more often; this is consistent with the observations of Somepalli et al. (2023b). Our theory also suggests why duplications increase memorization: the existence of a strong boundary between modes, which requires modes to have comparable probability mass, is degraded.

**Limitations of theoretical approach.** Our theory is simplified in at least two ways. First, only a simple formulation of training (via DSM) and sampling (via the PF-ODE) from diffusion models is considered. There exist alternatives to DSM, like sliced score matching (Song et al., 2020), and alternative ways of sampling, including using auxiliary momentum-like variables (Dockhorn et al., 2022b). Also, our theoretical analysis neglects variation due to numerical integration schemes, even though these may matter in practice (Liu et al., 2022; Karras et al., 2022; Dockhorn et al., 2022a).

Second, we study only unconditional models for simplicity. This means that in particular do not consider diffusion coupled to attention layers, which enables the text-conditioning behind many of the most striking diffusion-model-related successes (Rombach et al., 2022; Blattmann et al., 2023).

Finally, we do not consider realistic architectures (like U-nets) and rich learning dynamics due to theoretical tractability. However, these challenges are not unique to the current setting. Despite our contribution's simplicity, we hope that it nonetheless provides a foundation for others to more rigorously understand the inductive biases and generalization capabilities of diffusion models.

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

# Appendix

See `https://github.com/john-vastola/gtv-iclr25` for code that produces Fig. 1-3.

## Table of Contents

## A  OPTIMIZING OBJECTIVE REPRODUCES TRAINING DISTRIBUTION

In this appendix, we characterize the optima of the naive and DSM objectives introduced in Sec. 2, and in particular show that one (naively) theoretically expects diffusion models to reproduce the training distribution in the absence of expressivity-related constraints.

### A.1  DENOISING SCORE MATCHING PRESERVES OPTIMA OF NAIVE OBJECTIVE

First, we reestablish the well-known fact that the optima of the naive objective

$$J_0(\boldsymbol{\theta}) := \frac{1}{2} \, \mathbb{E}_{t,\boldsymbol{x}} \left\{ \lambda_t \| \hat{\boldsymbol{s}}_{\boldsymbol{\theta}}(\boldsymbol{x},t) - \boldsymbol{s}(\boldsymbol{x},t) \|_2^2 \right\} = \int \frac{\lambda_t}{2} \| \hat{\boldsymbol{s}}_{\boldsymbol{\theta}}(\boldsymbol{x},t) - \boldsymbol{s}(\boldsymbol{x},t) \|_2^2 \, p(\boldsymbol{x}|t)p(t) \, d\boldsymbol{x}dt \tag{19}$$

and DSM objective

$$J_1(\boldsymbol{\theta}) := \frac{1}{2} \mathbb{E}_{t,\boldsymbol{x}_0,\boldsymbol{x}} \left\{ \lambda_t \| \hat{\boldsymbol{s}}_{\boldsymbol{\theta}}(\boldsymbol{x},t) - \tilde{\boldsymbol{s}}(\boldsymbol{x},t;\boldsymbol{x}_0) \|_2^2 \right\}$$
$$= \int \frac{\lambda_t}{2} \| \hat{\boldsymbol{s}}_{\boldsymbol{\theta}}(\boldsymbol{x},t) - \tilde{\boldsymbol{s}}(\boldsymbol{x},t;\boldsymbol{x}_0) \|_2^2 \, p(\boldsymbol{x}|\boldsymbol{x}_0,t)p_{data}(\boldsymbol{x}_0)p(t) \, d\boldsymbol{x}d\boldsymbol{x}_0dt \tag{20}$$

are the same (Vincent, 2011; Song & Ermon, 2019; Song et al., 2021). Assume that $\boldsymbol{x},\boldsymbol{x}_0 \in \mathbb{R}^D$ and that $\boldsymbol{\theta} \in \mathbb{R}^F$. The gradient of $J_0$ with respect to $\boldsymbol{\theta}$ is

$$\frac{\partial J_0}{\partial \boldsymbol{\theta}} = \int \lambda_t \frac{\partial \hat{\boldsymbol{s}}_{\boldsymbol{\theta}}(\boldsymbol{x},t)^T}{\partial \boldsymbol{\theta}} \left[ \hat{\boldsymbol{s}}_{\boldsymbol{\theta}}(\boldsymbol{x},t) - \boldsymbol{s}(\boldsymbol{x},t) \right] \, p(\boldsymbol{x}|t)p(t) \, d\boldsymbol{x}dt \tag{21}$$

where $\partial \hat{\boldsymbol{s}}_{\boldsymbol{\theta}}(\boldsymbol{x},t)/\partial \boldsymbol{\theta}$ is the $D \times F$ Jacobian matrix of the score estimator. The gradient of $J_1$ is

$$\frac{\partial J_1}{\partial \boldsymbol{\theta}} = \int \lambda_t \frac{\partial \hat{\boldsymbol{s}}_{\boldsymbol{\theta}}(\boldsymbol{x},t)^T}{\partial \boldsymbol{\theta}} \left[ \hat{\boldsymbol{s}}_{\boldsymbol{\theta}}(\boldsymbol{x},t) - \tilde{\boldsymbol{s}}(\boldsymbol{x},t;\boldsymbol{x}_0) \right] \, p(\boldsymbol{x}|\boldsymbol{x}_0,t)p_{data}(\boldsymbol{x}_0)p(t) \, d\boldsymbol{x}d\boldsymbol{x}_0dt \ . \tag{22}$$

At this point, we make two observations about the gradient of $J_1$. First, the term on the left does not depend on $\boldsymbol{x}_0$, so we can marginalize over $\boldsymbol{x}_0$. Explicitly,

$$\int \lambda_t \frac{\partial \hat{\boldsymbol{s}}_{\boldsymbol{\theta}}(\boldsymbol{x},t)^T}{\partial \boldsymbol{\theta}} \hat{\boldsymbol{s}}_{\boldsymbol{\theta}}(\boldsymbol{x},t) \, p(\boldsymbol{x}|\boldsymbol{x}_0,t)p_{data}(\boldsymbol{x}_0)p(t) \, d\boldsymbol{x}d\boldsymbol{x}_0dt = \int \lambda_t \frac{\partial \hat{\boldsymbol{s}}_{\boldsymbol{\theta}}(\boldsymbol{x},t)^T}{\partial \boldsymbol{\theta}} \hat{\boldsymbol{s}}_{\boldsymbol{\theta}}(\boldsymbol{x},t) \, p(\boldsymbol{x}|t)p(t) \, d\boldsymbol{x}dt \ .$$

Second, the term on the right only depends on $\boldsymbol{x}_0$ through the proxy score target. Moreover,

$$\int \tilde{\boldsymbol{s}}(\boldsymbol{x},t;\boldsymbol{x}_0) \, p(\boldsymbol{x}|\boldsymbol{x}_0,t)p_{data}(\boldsymbol{x}_0) \, d\boldsymbol{x}_0 = \int \nabla_{\boldsymbol{x}} \log p(\boldsymbol{x}|\boldsymbol{x}_0,t) \, p(\boldsymbol{x}|\boldsymbol{x}_0,t)p_{data}(\boldsymbol{x}_0) \, d\boldsymbol{x}_0$$
$$= \int \nabla_{\boldsymbol{x}} p(\boldsymbol{x}|\boldsymbol{x}_0,t)p_{data}(\boldsymbol{x}_0) \, d\boldsymbol{x}_0$$
$$= \nabla_{\boldsymbol{x}} \int p(\boldsymbol{x}|\boldsymbol{x}_0,t)p_{data}(\boldsymbol{x}_0) \, d\boldsymbol{x}_0 \tag{23}$$
$$= \nabla_{\boldsymbol{x}} p(\boldsymbol{x}|t)$$
$$= \boldsymbol{s}(\boldsymbol{x},t)p(\boldsymbol{x}|t) \ .$$

Hence, the gradient of $J_0$ is the same as the gradient of $J_1$, so they have the same optima. If the score approximator is arbitrarily expressive and smooth in its parameters, we in particular have that the true score (a global minimum of $J_0$) is an optimum of the DSM objective.

This optimum is *also* the global minimum of $J_1$. Note that $J_1$ can be written as

$$\mathbb{E}_{t,\boldsymbol{x}_0,\boldsymbol{x}} \left\{ \frac{\lambda_t}{2} \| \hat{\boldsymbol{s}}_{\boldsymbol{\theta}}(\boldsymbol{x},t) - \boldsymbol{s}(\boldsymbol{x},t) + \boldsymbol{s}(\boldsymbol{x},t) - \tilde{\boldsymbol{s}}(\boldsymbol{x},t;\boldsymbol{x}_0) \|_2^2 \right\}$$

$$= \mathbb{E}_{t,\boldsymbol{x}_0,\boldsymbol{x}} \left\{ \frac{\lambda_t}{2} \left( \| \hat{\boldsymbol{s}}_{\boldsymbol{\theta}}(\boldsymbol{x},t) - \boldsymbol{s}(\boldsymbol{x},t) \|_2^2 + 2[\hat{\boldsymbol{s}}_{\boldsymbol{\theta}}(\boldsymbol{x},t) - \boldsymbol{s}(\boldsymbol{x},t)] \cdot [\boldsymbol{s}(\boldsymbol{x},t) - \tilde{\boldsymbol{s}}(\boldsymbol{x},t;\boldsymbol{x}_0)] + \| \boldsymbol{s}(\boldsymbol{x},t) - \tilde{\boldsymbol{s}}(\boldsymbol{x},t;\boldsymbol{x}_0) \|_2^2 \right) \right\} \ .$$

The first term is precisely equal to $J_0$. The second term vanishes, since (as shown by Eq. 23)

$$\mathbb{E}_{\boldsymbol{x}_0|\boldsymbol{x},t}[\tilde{\boldsymbol{s}}(\boldsymbol{x},t;\boldsymbol{x}_0)] = \boldsymbol{s}(\boldsymbol{x},t) \ . \tag{24}$$

Hence, we have that

$$J_1 = J_0 + \frac{1}{2} \mathbb{E}_{t,\boldsymbol{x}} \left\{ \lambda_t \, \text{tr}( \, \text{Cov}_{\boldsymbol{x}_0|\boldsymbol{x},t}(\tilde{\boldsymbol{s}}) \, ) \right\} \ . \tag{25}$$

In words: $J_1$ is equal to $J_0$ up to a $\boldsymbol{\theta}$-independent term that is a weighted combination of proxy score variances.

## A.2 TRAINING DISTRIBUTION REPRODUCTION

In practice, the training set consists of $1 \leq M < \infty$ examples (e.g., images) which together define

$$p_{data}(\boldsymbol{x}_0) = \frac{1}{M} \sum_{m=1}^{M} \delta(\boldsymbol{x}_0 - \boldsymbol{\mu}_m) . \tag{26}$$

The corresponding 'corrupted' distribution, given our choice of forward process (see Sec. 2), is

$$p(\boldsymbol{x}|t) = \frac{1}{M} \sum_{m=1}^{M} \mathcal{N}(\boldsymbol{x}; \alpha_t \boldsymbol{\mu}_m, \boldsymbol{S}_t) . \tag{27}$$

Usually, model updates utilize batches of samples from $p(\boldsymbol{x}, \boldsymbol{x}_0, t)$ (Song et al., 2021; Karras et al., 2022). As training proceeds, the model sees an ever larger number $P$ of samples from this distribution, making the empirical objective

$$J_1(\boldsymbol{\theta}; P) := \frac{1}{P} \sum_{n=1}^{P} \frac{\lambda(t^{(n)})}{2} \|\hat{\boldsymbol{s}}_{\boldsymbol{\theta}}(\boldsymbol{x}^{(n)}, t^{(n)}) - \tilde{\boldsymbol{s}}(\boldsymbol{x}^{(n)}, t^{(n)}; \boldsymbol{x}_0^{(n)})\|_2^2 , \tag{28}$$

where the $n$ superscripts index different (independent) samples from $p(\boldsymbol{x}, \boldsymbol{x}_0, t) = p(\boldsymbol{x}|\boldsymbol{x}_0, t)p_{data}(\boldsymbol{x}_0)p(t)$. For $P$ sufficiently large, by the central limit theorem, we expect the empirical objective to be extremely close to the true objective, and hence share its global minimum. But the global minimum is the true score, i.e.,

$$\boldsymbol{s}(\boldsymbol{x}, t) = \sum_{m=1}^{M} \boldsymbol{S}_t^{-1}(\alpha_t \boldsymbol{\mu}_m - \boldsymbol{x}) \frac{\mathcal{N}(\boldsymbol{x}; \alpha_t \boldsymbol{\mu}_m, \boldsymbol{S}_t)}{\sum_{m'} \mathcal{N}(\boldsymbol{x}; \alpha_t \boldsymbol{\mu}_{m'}, \boldsymbol{S}_t)} .$$

Since integrating the PF-ODE using this score produces samples from $p_{data}(\boldsymbol{x}_0)$—as $t \to 0$, $\boldsymbol{S}_t \to \boldsymbol{0}_D$, so the asymptotic 'force' pushing $\boldsymbol{x}_t$ towards an example becomes infinitely strong—we expect expressive diffusion models trained on the DSM objective using a large number of samples to reproduce training examples.

## B COVARIANCE OF PROXY SCORE

In this appendix, we compute the covariance of the proxy score $\tilde{s}(\boldsymbol{x}, t; \boldsymbol{x}_0) := \nabla_{\boldsymbol{x}} \log p(\boldsymbol{x}|\boldsymbol{x}_0, t)$ with respect to $p(\boldsymbol{x}_0|\boldsymbol{x}, t)$. We also show how this covariance is connected to Fisher information, and explicitly compute it in the case that $p_{data}(\boldsymbol{x}_0)$ is an isotropic Gaussian mixture.

### B.1 COMPUTING COVARIANCE OF PROXY SCORE

Note that

$$\frac{\partial^2}{\partial x_i \partial x_j} p(\boldsymbol{x}|\boldsymbol{x}_0, t) = \left[ -S_{t,ij}^{-1} + \tilde{s}_i \tilde{s}_j \right] p(\boldsymbol{x}|\boldsymbol{x}_0, t) . \tag{29}$$

Using this fact, we can write

$$
\begin{aligned}
\mathrm{Cov}_{\boldsymbol{x}_0|\boldsymbol{x},t}(\tilde{s}_i, \tilde{s}_j) &= \int \tilde{s}_i \tilde{s}_j \frac{p(\boldsymbol{x}|\boldsymbol{x}_0,t)p_{data}(\boldsymbol{x}_0)}{p(\boldsymbol{x}|t)} d\boldsymbol{x}_0 - s_i s_j \\
&= \int \frac{1}{p(\boldsymbol{x}|t)} \left[ S_{t,ij}^{-1} + \frac{\partial^2}{\partial x_i \partial x_j} \right] p(\boldsymbol{x}|\boldsymbol{x}_0,t)p_{data}(\boldsymbol{x}_0) \, d\boldsymbol{x}_0 - s_i s_j \\
&= \int \frac{1}{p(\boldsymbol{x}|t)} \left[ S_{t,ij}^{-1} + \frac{\partial^2}{\partial x_i \partial x_j} \right] p(\boldsymbol{x}|t) \frac{p(\boldsymbol{x}|\boldsymbol{x}_0,t)p_{data}(\boldsymbol{x}_0)}{p(\boldsymbol{x}|t)} \, d\boldsymbol{x}_0 - s_i s_j \quad (30) \\
&= S_{t,ij}^{-1} + \frac{1}{p(\boldsymbol{x}|t)} \frac{\partial^2 p(\boldsymbol{x}|t)}{\partial x_i \partial x_j} - s_i s_j \\
&= S_{t,ij}^{-1} + \frac{\partial^2}{\partial x_i \partial x_j} \log p(\boldsymbol{x}|t) .
\end{aligned}
$$

### B.2 CONNECTION TO FISHER INFORMATION

By definition, if $p(\boldsymbol{x}_0|\boldsymbol{x}, t)$ is viewed as a distribution with parameter vector $\boldsymbol{x}$, and $t$ is viewed as a hyperparameter, the Fisher information $\mathcal{I}_F$ is defined as

$$
\begin{aligned}
\mathcal{I}_F(\boldsymbol{x}|t) &:= \int \frac{\partial \log p(\boldsymbol{x}_0|\boldsymbol{x},t)}{\partial x_i} \cdot \frac{\partial \log p(\boldsymbol{x}_0|\boldsymbol{x},t)}{\partial x_j} p(\boldsymbol{x}_0|\boldsymbol{x},t) \, d\boldsymbol{x}_0 \\
&= \int \left[ \frac{\partial \log p(\boldsymbol{x}|\boldsymbol{x}_0,t)}{\partial x_i} - \frac{\partial \log p(\boldsymbol{x}|t)}{\partial x_i} \right] \left[ \frac{\partial \log p(\boldsymbol{x}|\boldsymbol{x}_0,t)}{\partial x_j} - \frac{\partial \log p(\boldsymbol{x}|t)}{\partial x_j} \right] p(\boldsymbol{x}_0|\boldsymbol{x},t) \, d\boldsymbol{x}_0 \\
&\qquad\qquad (31) \\
&= \int [\tilde{s}_i - s_i] [\tilde{s}_j - s_j] \, p(\boldsymbol{x}_0|\boldsymbol{x},t) \, d\boldsymbol{x}_0 \\
&= \mathrm{Cov}_{\boldsymbol{x}_0|\boldsymbol{x},t} (\tilde{s}_i, \tilde{s}_j) .
\end{aligned}
$$

### B.3 EXPLICIT COVARIANCE FOR ISOTROPIC GAUSSIAN MIXTURE TRAINING DISTRIBUTION

Suppose that $p(\boldsymbol{x}_0)$ and $p(\boldsymbol{x}|t)$ are

$$p_{data}(\boldsymbol{x}_0) = \frac{1}{M} \sum_m \mathcal{N}(\boldsymbol{x}_0; \boldsymbol{\mu}_m, \sigma_0^2 \boldsymbol{I}) \qquad\qquad p(\boldsymbol{x}|t) = \frac{1}{M} \sum_m \mathcal{N}(\boldsymbol{x}; \alpha_t \boldsymbol{\mu}_m, \alpha_t^2 \sigma_0^2 \boldsymbol{I} + \boldsymbol{S}_t) . \tag{32}$$

Note that the delta mixture case is an example ($\sigma_0^2 = 0$). Define the softmax distribution

$$p(m|\boldsymbol{x},t) := \frac{\mathcal{N}(\boldsymbol{x}; \alpha_t \boldsymbol{\mu}_m, \alpha_t^2 \sigma_0^2 \boldsymbol{I} + \boldsymbol{S}_t)}{\sum_{m'} \mathcal{N}(\boldsymbol{x}; \alpha_t \boldsymbol{\mu}_{m'}, \alpha_t^2 \sigma_0^2 \boldsymbol{I} + \boldsymbol{S}_t)} \tag{33}$$

on $\mathcal{M} = \{1, ..., M\}$. This distribution, whose moments determine the proxy score covariance, has a Bayesian interpretation: it corresponds to an ideal observer's belief about the outcome $x_0$, given that said observer is in state $x$ at time $t$.

The first and second derivatives of $p(\boldsymbol{x}|t)$ can be written in terms of expectations with respect to this distribution, since

$$\frac{1}{p(\boldsymbol{x}|t)} \frac{\partial p(\boldsymbol{x}|t)}{\partial \boldsymbol{x}} = \sum_m (\alpha_t^2 \sigma_0^2 \boldsymbol{I} + \boldsymbol{S}_t)^{-1}(\alpha_t \boldsymbol{\mu}_m - \boldsymbol{x})p(m|\boldsymbol{x},t) = (\alpha_t^2 \sigma_0^2 \boldsymbol{I} + \boldsymbol{S}_t)^{-1}(\alpha_t \langle \boldsymbol{\mu} \rangle_{\mathcal{M}} - \boldsymbol{x})$$

and the Hessian matrix $(H_{ij} := \partial_{ij}^2 p(\boldsymbol{x}|t))$ is

$$\frac{\boldsymbol{H}}{p(\boldsymbol{x}|t)} = \sum_m \left[ -(\alpha_t^2 \sigma_0^2 \boldsymbol{I} + \boldsymbol{S}_t)^{-1} + (\alpha_t^2 \sigma_0^2 \boldsymbol{I} + \boldsymbol{S}_t)^{-1} (\alpha_t \boldsymbol{\mu}_m - \boldsymbol{x})(\alpha_t \boldsymbol{\mu}_m - \boldsymbol{x})^T (\alpha_t^2 \sigma_0^2 \boldsymbol{I} + \boldsymbol{S}_t)^{-1} \right] p(m|\boldsymbol{x}, t)$$

$$= -(\alpha_t^2 \sigma_0^2 \boldsymbol{I} + \boldsymbol{S}_t)^{-1} + (\alpha_t^2 \sigma_0^2 \boldsymbol{I} + \boldsymbol{S}_t)^{-1} \mathbb{E}_{\mathcal{M}} \left\{ (\alpha_t \boldsymbol{\mu} - \boldsymbol{x})(\alpha_t \boldsymbol{\mu} - \boldsymbol{x})^T \right\} (\alpha_t^2 \sigma_0^2 \boldsymbol{I} + \boldsymbol{S}_t)^{-1}$$

$$= -(\alpha_t^2 \sigma_0^2 \boldsymbol{I} + \boldsymbol{S}_t)^{-1} + (\alpha_t^2 \sigma_0^2 \boldsymbol{I} + \boldsymbol{S}_t)^{-1} \left[ \alpha_t^2 \mathrm{Cov}_{\mathcal{M}}(\boldsymbol{\mu}) + (\alpha_t \langle \boldsymbol{\mu} \rangle_{\mathcal{M}} - \boldsymbol{x})(\alpha_t \langle \boldsymbol{\mu} \rangle_{\mathcal{M}} - \boldsymbol{x})^T \right] (\alpha_t^2 \sigma_0^2 \boldsymbol{I} + \boldsymbol{S}_t)^{-1} .$$

Then we have

$$\frac{\partial^2 \log p(\boldsymbol{x}|t)}{\partial x_i \partial x_j} = -(\alpha_t^2 \sigma_0^2 \boldsymbol{I} + \boldsymbol{S}_t)^{-1} + \alpha_t^2 (\alpha_t^2 \sigma_0^2 \boldsymbol{I} + \boldsymbol{S}_t)^{-1} \mathrm{Cov}_{\mathcal{M}}(\boldsymbol{\mu})(\alpha_t^2 \sigma_0^2 \boldsymbol{I} + \boldsymbol{S}_t)^{-1} \qquad (34)$$

and hence that

$$\mathrm{Cov}_{\boldsymbol{x}_0|\boldsymbol{x},t}(\tilde{\boldsymbol{s}}) = \boldsymbol{S}_t^{-1} - (\alpha_t^2 \sigma_0^2 \boldsymbol{I} + \boldsymbol{S}_t)^{-1} + \alpha_t^2 (\alpha_t^2 \sigma_0^2 \boldsymbol{I} + \boldsymbol{S}_t)^{-1} \mathrm{Cov}_{\mathcal{M}}(\boldsymbol{\mu})(\alpha_t^2 \sigma_0^2 \boldsymbol{I} + \boldsymbol{S}_t)^{-1} .$$

For a delta mixture training distribution, since $\sigma_0^2 = 0$, the covariance simplifies to

$$\mathrm{Cov}_{\boldsymbol{x}_0|\boldsymbol{x},t}(\tilde{\boldsymbol{s}}) = \alpha_t^2 \boldsymbol{S}_t^{-1} \mathrm{Cov}_{\mathcal{M}}(\boldsymbol{\mu}) \boldsymbol{S}_t^{-1} . \qquad (35)$$

The above equation implies that the covariance of the proxy score is, up to scaling, the same as uncertainty about $\boldsymbol{x}_0$ given $\boldsymbol{x}$ and $t$.

## C  BOUNDARY REGIONS: DEFINITION AND BAYESIAN INTERPRETATION

A key concept used throughout this paper is that of a *boundary region*, which we informally define as a region of $\mathbb{R}^D$ between two or more training examples in the case that $p_{data}$ is discrete. Intuitively, these regions correspond to the 'gaps' in the training distribution, and a reasonable generalization strategy is to fill them in.

In this appendix, we briefly comment that the notion of a boundary region can be made more precise via Bayes' theorem: for a given reverse diffusion time $t$, boundary regions are sets of $x$ values for which uncertainty about the endpoint $x_0$ is particularly high. For discrete data distributions, such regions coincide with sets of states between training examples, since one is maximally uncertain about the endpoint when $x$ is equidistant from two or more training examples.

An illustrative one-dimensional example involves two training examples at $x_0 = \pm\mu$. The noise-corrupted data distribution at time $t$ is

$$p(x|t) = \frac{1}{2}\mathcal{N}(x; \alpha_t\mu, \sigma_t^2) + \frac{1}{2}\mathcal{N}(x; -\alpha_t\mu, \sigma_t^2) \,, \tag{36}$$

so the posterior endpoint estimate given $x, t$ is the softmax distribution (see also Appendix B)

$$
\begin{aligned}
p(x_0|x,t) &= \frac{p(x|x_0,t)p_{data}(x_0)}{\sum_{x_0} p(x|x_0,t)p_{data}(x_0)} \\
&= \frac{\mathcal{N}(x; \alpha_t\mu, \sigma_t^2)}{\mathcal{N}(x; \alpha_t\mu, \sigma_t^2) + \mathcal{N}(x; -\alpha_t\mu, \sigma_t^2)}\delta(x_0 - \mu) + \frac{\mathcal{N}(x; -\alpha_t\mu, \sigma_t^2)}{\mathcal{N}(x; \alpha_t\mu, \sigma_t^2) + \mathcal{N}(x; -\alpha_t\mu, \sigma_t^2)}\delta(x_0 + \mu) \,.
\end{aligned}
$$

The mean $\mathbb{E}[x_0|x,t]$ of this distribution is

$$\mathbb{E}[x_0|x,t] = \sum_{x_0} x_0 \, p(x_0|x,t) = \mu \tanh\left(\frac{\alpha_t\mu}{\sigma_t^2}x\right) \,. \tag{37}$$

The interpretation of this quantity is interesting in light of the 'Bayesian guessing game' metaphor for score learning (see, e.g., Kamb & Ganguli (2024)). One imagines that one starts at an unknown $x_0$ (here, either $+\mu$ or $-\mu$), and then noise is added according to the forward process until time $t$. Given that an observer is in state $x$ at time $t$, what was the likely starting point $x_0$? The quantity $\mathbb{E}[x_0|x,t]$ is the Bayes-optimal solution to this problem.

In the context of this toy example, it has the following form. If $x$ is very positive, one tends to believe $x_0 = +\mu$; if $x$ is very negative, one tends to believe $x_0 = -\mu$. For intermediate $x$, especially near $x = 0$, uncertainty is highest, and $\mathbb{E}[x_0|x,t]$ is near zero, since the observer could have plausibly started at either $x_0 = +\mu$ or $x_0 = -\mu$.

This high uncertainty allows us to formalize the idea that the region between $+\mu$ and $-\mu$, especially near $x = 0$, is a boundary region. Quantitatively, we have

$$\text{var}(x_0|x,t) = \sum_{x_0} x_0^2 p(x_0|x,t) - \mathbb{E}[x_0|x,t]^2 = \mu^2\left[1 - \tanh^2\left(\frac{\alpha_t\mu}{\sigma_t^2}x\right)\right] = \frac{\mu^2}{\cosh^2\left(\frac{\alpha_t\mu}{\sigma_t^2}x\right)} \tag{38}$$

or equivalently

$$\sqrt{\text{var}(x_0|x,t)} = \frac{\mu}{\cosh\left(\frac{\alpha_t\mu}{\sigma_t^2}x\right)} \,. \tag{39}$$

At $x = 0$, the standard deviation of $p(x_0|x,t)$ equals $\mu$, i.e., there is maximum uncertainty about the starting point $x_0$. Moreover, it is fairly high until $x \approx \frac{\sigma_t^2}{\alpha_t\mu}$, which also shows that the effective size of a boundary region is smaller at smaller noise scales. Said differently, the basins of attraction surrounding each training example become increasingly sharp as $\sigma_t \to 0$.

## D  PATH-INTEGRAL REPRESENTATION OF LEARNED DISTRIBUTION

In this appendix, we derive a path-integral description of the 'typical' distribution learned by diffusion models. We do this in three stages. First, we derive a path-integral description of the PF-ODE. Next, we derive a path-integral description of a more general kind of stochastic process. Finally, we show that averaging the path-integral representation of the PF-ODE over sample realizations produces a path integral whose dynamics correspond to those of the aforementioned stochastic process.

### D.1  WARM-UP: DERIVING A PATH-INTEGRAL REPRESENTATION OF THE PF-ODE

A general ODE can be written as

$$\dot{\boldsymbol{x}}_t = \boldsymbol{f}(\boldsymbol{x}_t, t) \tag{40}$$

where $\boldsymbol{x}_t \in \mathbb{R}^D$ and $t \in [\epsilon, T]$. We will assume that $\boldsymbol{f}$ is smooth to avoid technical issues. If we discretize time, and slightly abuse notation by using $t$ and $T$ to refer to integer-valued indices instead of real-valued times, we can write the trajectory as $\{x_T, x_{T-1}, ..., x_1, x_0\}$ and the corresponding updates in the form

$$\boldsymbol{x}_t = \boldsymbol{x}_{t+1} - \boldsymbol{f}(\boldsymbol{x}_{t+1}, t+1)\Delta t . \tag{41}$$

Note that our discretization corresponds to a first-order Euler update scheme. In the small $\Delta t$ limit, this specific choice does not matter, even if it matters in practice; we use it to slightly simplify our argument. Conditional on the initial point $\boldsymbol{x}_T$, the probability of reaching another point $\boldsymbol{x}_0$ after $T$ backwards-time steps is

$$p(\boldsymbol{x}_0|\boldsymbol{x}_T) = \int \delta(\boldsymbol{x}_0 - \boldsymbol{x}_1 + \boldsymbol{f}(\boldsymbol{x}_1, 1)\Delta t) \cdots \delta(\boldsymbol{x}_{T-1} - \boldsymbol{x}_T + \boldsymbol{f}(\boldsymbol{x}_T, T)\Delta t) \, d\boldsymbol{x}_1 \cdots d\boldsymbol{x}_{T-1} \tag{42}$$

where $\delta$ is the Dirac delta function. Here, we will employ a well-known integral representation of the Dirac delta function:

$$\delta(\boldsymbol{x} - \boldsymbol{x}') = \int \frac{d\boldsymbol{p}}{(2\pi)^D} \, \exp\left\{-i\boldsymbol{p} \cdot (\boldsymbol{x} - \boldsymbol{x}')\right\} \tag{43}$$

where $\boldsymbol{p}$ is integrated over all of $\mathbb{R}^D$. Our expression for $p(\boldsymbol{x}_0|\boldsymbol{x}_T)$ becomes

$$p(\boldsymbol{x}_0|\boldsymbol{x}_T) = \int \frac{d\boldsymbol{p}_0}{(2\pi)^D} \frac{d\boldsymbol{x}_1 d\boldsymbol{p}_1}{(2\pi)^D} \cdots \frac{d\boldsymbol{x}_{T-1} d\boldsymbol{p}_{T-1}}{(2\pi)^D} \, \exp\left\{\sum_{t=0}^{T-1} -i\boldsymbol{p}_t \cdot [\boldsymbol{x}_t - \boldsymbol{x}_{t+1} + \boldsymbol{f}(\boldsymbol{x}_{t+1}, t+1)\Delta t]\right\} . \tag{44}$$

Schematically, we can write this path integral as a 'sum over paths'

$$p(\boldsymbol{x}_0|\boldsymbol{x}_T) = \int \mathcal{D}[\boldsymbol{p}_t]\mathcal{D}[\boldsymbol{x}_t] \, \exp\left\{\int_\epsilon^T -i\boldsymbol{p}_t \cdot [-\dot{\boldsymbol{x}}_t + \boldsymbol{f}(\boldsymbol{x}_t, t)] \, dt\right\} , \tag{45}$$

although explicitly using this form is unnecessary for our purposes. (This is good, since remaining in discrete time allows us to avoid various thorny mathematical issues.) For the particular choice of $\boldsymbol{f}$ associated with the PF-ODE, we have discrete and schematic forms

$$p(\boldsymbol{x}_0|\boldsymbol{x}_T) = \int \frac{d\boldsymbol{p}_0}{(2\pi)^D} \frac{d\boldsymbol{x}_1 d\boldsymbol{p}_1}{(2\pi)^D} \cdots \frac{d\boldsymbol{x}_{T-1} d\boldsymbol{p}_{T-1}}{(2\pi)^D} \, e^{\sum_{t=0}^{T-1} -i\boldsymbol{p}_t \cdot [\boldsymbol{x}_t - \boldsymbol{x}_{t+1} - (\beta_{t+1}\boldsymbol{x}_{t+1} + \boldsymbol{D}_{t+1}\boldsymbol{s}(\boldsymbol{x}_{t+1}, t+1))\Delta t]}$$

$$p(\boldsymbol{x}_0|\boldsymbol{x}_T) = \int \mathcal{D}[\boldsymbol{p}_t]\mathcal{D}[\boldsymbol{x}_t] \, \exp\left\{\int_\epsilon^T i\boldsymbol{p}_t \cdot [\dot{\boldsymbol{x}}_t + \beta_t\boldsymbol{x}_t + \boldsymbol{D}_t\boldsymbol{s}(\boldsymbol{x}_t, t)] \, dt\right\} .$$

## D.2 Deriving a path-integral representation of a more general process

Consider a more general type of backwards, discrete-time stochastic process. Once again, suppose that a variable $\boldsymbol{x}_t \in \mathbb{R}^D$ evolves backwards in time from an initial point $\boldsymbol{x}_T$. But this time, suppose that the transition between $\boldsymbol{x}_{t+1}$ and $\boldsymbol{x}_t$ depends upon some set of $K$ independent standard normal random variables $\{\xi_k\}$. In particular, suppose that discrete-time updates have the form

$$x_{tj} = x_{t+1,j} - f_j(\boldsymbol{x}_{t+1}, t+1)\Delta t + \sum_{k=1}^{K} G_{jk}(\boldsymbol{x}_{t+1}, t+1)\,\xi_k\,\Delta t\,, \tag{46}$$

i.e., updates are the same as before except for the new noise term. In general, the noise term is quite complicated; $\boldsymbol{G}$ is a $D \times K$ matrix which can depend explicitly on both the current state and the current time. The process described by the above updates is generally not Markov, since noise added at different time steps can depend on some of the same $\xi_k$ variables, and hence the amount of noise added at one time step can be correlated with the amount of noise added at some other time step.

What is the distribution of $\boldsymbol{x}_0$, the result of $T$ steps of this process, conditional on a starting point $\boldsymbol{x}_T$? We know that each update depends only on the previous state and the noise variables, so

$$p(\boldsymbol{x}_0|\boldsymbol{x}_T) = \int p(\boldsymbol{x}_0|\boldsymbol{x}_1, \{\xi_k\})p(\boldsymbol{x}_1|\boldsymbol{x}_2, \{\xi_k\}) \cdots p(\boldsymbol{x}_{T-1}|\boldsymbol{x}_T, \{\xi_k\})\, p(\{\xi_k\})\, d\boldsymbol{x}_1 \cdots d\boldsymbol{x}_{T-1}d\{\xi_k\}\,.$$

In particular, conditional on the previous state and the noise variables, updates are deterministic. This allows us to write the above transition probability as

$$\int \left[ \prod_{j=1}^{D} \prod_{t=0}^{T-1} \delta\left( x_{t,j} - x_{t+1,j} + f_j(\boldsymbol{x}_{t+1}, t+1)\Delta t + \sum_{k=1}^{K} G_{jk}(\boldsymbol{x}_{t+1}, t+1)\,\xi_k\,\Delta t \right) \right] p(\{\xi_k\})\, d\boldsymbol{x}_1 \cdots d\boldsymbol{x}_{T-1}d\{\xi_k\}\,.$$

Using the same integral representation of the Dirac delta function that we used above, this becomes

$$\int e^{\sum_{t,j} -ip_{t,j}\left[x_{t,j}-x_{t+1,j}+f_j(\boldsymbol{x}_{t+1},t+1)\Delta t+\sum_{k=1}^{K} G_{jk}(\boldsymbol{x}_{t+1},t+1)\,\xi_k\,\Delta t\right]} p(\{\xi_k\})\, \frac{d\boldsymbol{p}_0}{(2\pi)^D}\frac{d\boldsymbol{x}_1 d\boldsymbol{p}_1}{(2\pi)^D} \cdots \frac{d\boldsymbol{x}_{T-1}d\boldsymbol{p}_{T-1}}{(2\pi)^D}d\{\xi_k\}\,.$$

Although this appears to be extremely complicated, it can be considerably simplified by doing the integral over the noise variables. Since the noise variables are all independent and standard normal,

$$p(\{\xi_k\}) = \frac{1}{(2\pi)^{k/2}} \exp\left\{ -\frac{\xi_1^2}{2} - \cdots - \frac{\xi_K^2}{2} \right\}\,. \tag{47}$$

Hence, the integral over the noise variables is a typical Gaussian integral with a linear term. We can save time by recognizing the integral as essentially computing the characteristic function of a standard normal; more precisely, we have

$$\begin{aligned}
I_k &= \int \exp\left\{ -i\xi_k \sum_{t=0}^{T-1} \sum_{j=1}^{D} p_{t,j} G_{jk}(\boldsymbol{x}_{t+1}, t+1)\Delta t \right\} \frac{e^{-\xi_k^2/2}}{\sqrt{2\pi}}\, d\xi_k \\
&= \exp\left\{ -\frac{1}{2} \sum_{t=0}^{T-1} \sum_{t'=0}^{T-1} \sum_{j=1}^{D} \sum_{j'=1}^{D} p_{t,j} G_{jk}(\boldsymbol{x}_{t+1}, t+1) G_{j'k}(\boldsymbol{x}_{t'+1}, t'+1) p_{t',j'} \Delta t \Delta t \right\}
\end{aligned} \tag{48}$$

for each $\xi_k$. Putting everything together, we find that $p(\boldsymbol{x}_0|\boldsymbol{x}_T)$ can be written

$$\int e^{\sum_{t,j} -ip_{t,j}[x_{t,j}-x_{t+1,j}+f_j(\boldsymbol{x}_{t+1},t+1)\Delta t]-\frac{1}{2}\sum_{t,t',j,j'}\sum_{k=1}^{K} p_{t,j}G_{jk}(\boldsymbol{x}_{t+1},t+1)G_{j'k}(\boldsymbol{x}_{t'+1},t'+1)p_{t',j'}\Delta t\Delta t}\, \frac{d\{\boldsymbol{x}_t\}d\{\boldsymbol{p}_t\}}{(2\pi)^{DT}}$$

where we have used the shorthand $d\{\boldsymbol{x}_t\}d\{\boldsymbol{p}_t\} := d\boldsymbol{p}_0\, d\boldsymbol{x}_1 d\boldsymbol{p}_1 \cdots d\boldsymbol{x}_{T-1}d\boldsymbol{p}_{T-1}$. This is our final answer, although it is more enlightening to write it in its schematic continuous-time form. We obtain

$$p(\boldsymbol{x}_0|\boldsymbol{x}_T) = \int \mathcal{D}[\boldsymbol{p}_t]\mathcal{D}[\boldsymbol{x}_t] \exp\left\{ \int_\epsilon^T -i\boldsymbol{p}_t \cdot [-\dot{\boldsymbol{x}}_t + \boldsymbol{f}(\boldsymbol{x}_t, t)]\, dt - \frac{1}{2} \int_\epsilon^T \int_\epsilon^T \boldsymbol{p}_t^T \boldsymbol{V}(\boldsymbol{x}_t, t; \boldsymbol{x}_{t'}, t')\boldsymbol{p}_{t'}\, dt dt' \right\}$$

where we have defined the state- and time-dependent $D \times D$ V-kernel $V_{ij}(\boldsymbol{x}_t, t; \boldsymbol{x}_{t'}, t')$ via

$$V_{ij}(\boldsymbol{x}_t, t; \boldsymbol{x}_{t'}, t') := \sum_{k=1}^{K} G_{ik}(\boldsymbol{x}_t, t) G_{jk}(\boldsymbol{x}_{t'}, t')\,, \tag{49}$$

or equivalently via $\boldsymbol{V}(\boldsymbol{x}_t, t; \boldsymbol{x}_{t'}, t') := \boldsymbol{G}(\boldsymbol{x}_t, t)\,\boldsymbol{G}^T(\boldsymbol{x}_{t'}, t')$. Note that it is positive semidefinite.

### D.3 AVERAGING LEARNED DISTRIBUTION OVER SAMPLE REALIZATIONS

What is the 'typical' distribution learned by an ensemble of diffusion models which differ only in the samples each used during training? In this subsection, we show that the net effect of averaging over sample realizations is to contribute a noise term to the PF-ODE. The path-integral representation we obtain is of the class we discussed in the previous subsection.

Suppose a diffusion model is associated with a parameterized score approximator $\hat{\boldsymbol{s}}_{\boldsymbol{\theta}}(\boldsymbol{x}, t)$. The distribution learned by the diffusion model is then

$$q(\boldsymbol{x}_0|\boldsymbol{x}_T; \boldsymbol{\theta}) = \int \mathcal{D}[\boldsymbol{p}_t]\mathcal{D}[\boldsymbol{x}_t] \exp\left\{\int_{\epsilon}^{T} i\boldsymbol{p}_t \cdot [\dot{\boldsymbol{x}}_t + \beta_t \boldsymbol{x}_t + \boldsymbol{D}_t \hat{\boldsymbol{s}}_{\boldsymbol{\theta}}(\boldsymbol{x}_t, t)] \ dt\right\}, \tag{50}$$

where we have used the schematic form of the PF-ODE path-integral representation for clarity. (Moving to discrete time does not affect our arguments, but only makes notation more cumbersome.) Averaging over sample realizations is mathematically equivalent to computing the characteristic function of the score approximator. The sample-averaged $q$, $\mathbb{E}_{\boldsymbol{\theta}}[q(\boldsymbol{x}_0|\boldsymbol{x}_T; \boldsymbol{\theta})] = [q(\boldsymbol{x}_0|\boldsymbol{x}_T)]$, is

$$[q(\boldsymbol{x}_0|\boldsymbol{x}_T)] = \int \mathcal{D}[\boldsymbol{p}_t]\mathcal{D}[\boldsymbol{x}_t] \exp\left\{\int_{\epsilon}^{T} i\boldsymbol{p}_t \cdot [\dot{\boldsymbol{x}}_t + \beta_t \boldsymbol{x}_t] \ dt\right\} \mathbb{E}_{\boldsymbol{\theta}}\left[e^{\int_{\epsilon}^{T} i\boldsymbol{p}_t^T \boldsymbol{D}_t \hat{\boldsymbol{s}}_{\boldsymbol{\theta}}(\boldsymbol{x}_t, t) \ dt}\right]. \tag{51}$$

Assuming the score approximator ensemble is well-behaved, its characteristic function can be written as a cumulant expansion. Here, we have

$$\log \mathbb{E}_{\boldsymbol{\theta}}\left[e^{\int_{\epsilon}^{T} i\boldsymbol{p}_t^T \boldsymbol{D}_t \hat{\boldsymbol{s}}_{\boldsymbol{\theta}}(\boldsymbol{x}_t, t) \ dt}\right]$$
$$= \int_{\epsilon}^{T} i\boldsymbol{p}_t \boldsymbol{D}_t [\hat{\boldsymbol{s}}_{\boldsymbol{\theta}}(\boldsymbol{x}_t, t)] \ dt - \frac{1}{2} \int_{\epsilon}^{T} \int_{\epsilon}^{T} \boldsymbol{p}_t^T \boldsymbol{D}_t \text{Cov}_{\boldsymbol{\theta}}\left[\hat{\boldsymbol{s}}_{\boldsymbol{\theta}}(\boldsymbol{x}_t, t), \hat{\boldsymbol{s}}_{\boldsymbol{\theta}}(\boldsymbol{x}_{t'}, t')\right] \boldsymbol{D}_{t'} \boldsymbol{p}_{t'} + \cdots \tag{52}$$

where the dots indicate higher-order cumulants and $[\hat{\boldsymbol{s}}_{\boldsymbol{\theta}}(\boldsymbol{x}_t, t)]$ indicates the ensemble-averaged score approximator. In this work, we neglect the higher-order terms. Often, they are suppressed by some factor (e.g., the number of model parameters divided by the number of samples).

We obtain dynamics of the class described in the previous subsection. Here, the $D \times D$ V-kernel is

$$V_{ij}(\boldsymbol{x}_t, t; \boldsymbol{x}_{t'}, t') := \sum_{a,b} D_{t,ia} \text{Cov}_{\boldsymbol{\theta}}[\hat{s}_a(\boldsymbol{x}_t, t), \hat{s}_b(\boldsymbol{x}_{t'}, t')] D_{t',bj}, \tag{53}$$

or equivalently $\boldsymbol{V}(\boldsymbol{x}_t, t; \boldsymbol{x}_{t'}, t') := \boldsymbol{D}_t \text{Cov}_{\boldsymbol{\theta}}\left[\hat{\boldsymbol{s}}_{\boldsymbol{\theta}}(\boldsymbol{x}_t, t), \hat{\boldsymbol{s}}_{\boldsymbol{\theta}}(\boldsymbol{x}_{t'}, t')\right] \boldsymbol{D}_{t'}$.

# E    NAIVE SCORE ESTIMATORS GENERALIZE: DETAILS

In this appendix, we show that integrating the PF-ODE using naive score estimates yields a specific kind of generalization (Prop. 4.1). Suppose that we are integrating the PF-ODE from some initial point $\boldsymbol{x}_T$ using $T$ first-order Euler updates (or some other integration scheme; the choice does not matter in the continuous-time limit), so that

$$\boldsymbol{x}_t = \boldsymbol{x}_{t+1} + (\beta_{t+1}\boldsymbol{x}_{t+1} + \boldsymbol{D}_{t+1}\boldsymbol{s}(\boldsymbol{x}_{t+1}, t+1))\,\Delta t . \tag{54}$$

But suppose that we do not use the score function directly in our updates, but at each time step construct a noisy version of it based on a sample $\boldsymbol{x}_{0t} \sim p(\boldsymbol{x}_0|\boldsymbol{x}, t)$. In particular, consider the naive score estimator

$$\hat{\boldsymbol{s}}(\boldsymbol{x}_t, t) := \boldsymbol{s}(\boldsymbol{x}_t, t) + \sqrt{\frac{\kappa}{\Delta t}}[\tilde{\boldsymbol{s}}(\boldsymbol{x}_t, t; \boldsymbol{x}_{0t}) - \boldsymbol{s}(\boldsymbol{x}_t, t)] \tag{55}$$

where $\kappa \geq 0$ is a constant that controls the estimator's variance. Note that a new, independent sample $\boldsymbol{x}_{0t'}$ is drawn at each step $t'$. We are interested in studying the extent to which this scheme produces a distribution different from $p_{data}(\boldsymbol{x}_0)$.

Using the result from Appendix D, the typical learned distribution $[q(\boldsymbol{x}_0|\boldsymbol{x}_T)]$ is (approximately) characterized by the average and V-kernel of $\hat{\boldsymbol{s}}$. Since $\mathbb{E}_{\boldsymbol{x}_0}[\tilde{\boldsymbol{s}}(\boldsymbol{x}, t; \boldsymbol{x}_0)] = \boldsymbol{s}(\boldsymbol{x}, t)$, this score estimator is unbiased, i.e., $[\hat{\boldsymbol{s}}] = \boldsymbol{s}$. The V-kernel $\boldsymbol{V}(\boldsymbol{x}_t, t; \boldsymbol{x}_{t'}, t')$ is

$$\boldsymbol{V} := \boldsymbol{D}_t \text{Cov}_{\boldsymbol{\theta}}[\hat{\boldsymbol{s}}(\boldsymbol{x}_t, t), \hat{\boldsymbol{s}}(\boldsymbol{x}_{t'}, t')]\boldsymbol{D}_{t'} = \boldsymbol{D}_t \text{Cov}_{\boldsymbol{\theta}}[\hat{\boldsymbol{s}}(\boldsymbol{x}_t, t), \hat{\boldsymbol{s}}(\boldsymbol{x}_t, t)]\boldsymbol{D}_t\,\delta(t - t')\Delta t \tag{56}$$

since samples generated at different time steps are independent of one another. Moreover,

$$\text{Cov}_{\boldsymbol{\theta}}[\hat{\boldsymbol{s}}(\boldsymbol{x}_t, t)] = \frac{\kappa}{\Delta t}\text{Cov}_{\boldsymbol{\theta}}[\tilde{\boldsymbol{s}}(\boldsymbol{x}_t, t)] \tag{57}$$

since the only random part of the estimator is the proxy score. Finally,

$$\begin{aligned} V_{ij}(\boldsymbol{x}_t, t; \boldsymbol{x}_{t'}, t') &= \kappa \sum_{a,b} D_{t,ia}\text{Cov}_{\boldsymbol{\theta}}[\tilde{s}_a(\boldsymbol{x}_t, t), \tilde{s}_b(\boldsymbol{x}_t, t)]D_{t,bj}\,\delta(t - t') \\ &= \kappa \sum_{a,b} D_{t,ia}\left[S_{t,ab}^{-1} + \partial_{ab}^2 \log p(\boldsymbol{x}_t|t)\right]D_{t,bj}\,\delta(t - t') . \end{aligned} \tag{58}$$

Using $\boldsymbol{C}(\boldsymbol{x}, t)$ as shorthand for the proxy score covariance matrix, we equivalently have

$$\boldsymbol{V}(\boldsymbol{x}_t, t; \boldsymbol{x}_{t'}, t') = \kappa\boldsymbol{D}_t\boldsymbol{C}(\boldsymbol{x}_t, t)\boldsymbol{D}_t\,\delta(t - t') . \tag{59}$$

As a final technical note, note that the naive estimator must scale like $1/\sqrt{\Delta t}$ in order for the V-kernel to be nontrivial in the $\Delta t \to 0$ limit (and indeed, for the continuous-time limit to make sense). This is easiest to see in discrete time: since samples generated at different time steps $k$ and $\ell$ are independent, the V-kernel picks up a factor $\delta_{k\ell}$, which equals one when $k = \ell$ and is zero otherwise. In continuous time, this looks like $\delta(t - t')\Delta t$, *not* $\delta(t - t')$ (Eq. 56). This problematic $\Delta t$ factor can be canceled by a corresponding $1/(\Delta t)$ factor in the estimator covariance, which motivates making the estimator scale like $1/\sqrt{\Delta t}$.

# F  LINEAR SCORE ESTIMATOR: DETAILS

In this appendix, we compute the sample-realization-averaged distribution learned by a linear score estimator (Prop. 5.1). Whether it generalizes or not depends strongly on whether the number of features $F$ scales with the number of samples $P$ used during training. First, we must compute the optimum of the DSM objective for a linear model. Then we will determine the average and V-kernel of the optimal linear score estimator.

## F.1  DEFINITION OF LINEAR SCORE MODEL

Consider a linear score estimator

$$\hat{\boldsymbol{s}}_{\boldsymbol{\theta}}(\boldsymbol{x}, t) = \boldsymbol{w}_0 + \boldsymbol{W}\boldsymbol{\phi}(\boldsymbol{x}, t) \qquad \hat{s}_i(\boldsymbol{x}, t) = w_{0i} + \sum_{j=1}^{F} W_{ij}\phi_j(\boldsymbol{x}, t) , \qquad (60)$$

where the feature maps $\boldsymbol{\phi} = (\phi_1, ..., \phi_F)^T$ are linearly independent, smooth functions from $\mathbb{R}^D \times [0, T]$ to $\mathbb{R}$ that are square-integrable with respect to the measure $\lambda_t p(\boldsymbol{x}, t)$ for all $t$. The parameters to be estimated are $\boldsymbol{\theta} := \{\boldsymbol{w}_0, \boldsymbol{W}\}$, with $\boldsymbol{w}_0 \in \mathbb{R}^D$ and $\boldsymbol{W} \in \mathbb{R}^{D \times F}$.

## F.2  OPTIMUM OF DSM OBJECTIVE FOR LINEAR SCORE MODEL

For this estimator, the DSM objective reads

$$J_1(\boldsymbol{\theta}) = \int \frac{\lambda_t}{2} \|\boldsymbol{w}_0 + \boldsymbol{W}\boldsymbol{\phi}(\boldsymbol{x}, t) - \tilde{\boldsymbol{s}}(\boldsymbol{x}, t; \boldsymbol{x}_0)\|_2^2 \, p(\boldsymbol{x}|\boldsymbol{x}_0, t) p_{data}(\boldsymbol{x}_0) p(t) \, d\boldsymbol{x} d\boldsymbol{x}_0 dt . \qquad (61)$$

Note,

$$\frac{\partial \hat{s}_i}{\partial w_{0a}} = \delta_{ia} \qquad \frac{\partial \hat{s}_i}{\partial W_{ab}} = \delta_{ia}\phi_b . \qquad (62)$$

Using these to take the gradient of the DSM objective, we have

$$\frac{\partial J_1}{\partial w_{0a}} = \mathbb{E}_{\boldsymbol{x}, \boldsymbol{x}_0, t}\left\{ \lambda_t \left[ w_{0a} + \sum_{j=1}^{F} W_{aj}\phi_j(\boldsymbol{x}, t) - \tilde{s}_a(\boldsymbol{x}, t; \boldsymbol{x}_0) \right] \right\}$$

$$\frac{\partial J_1}{\partial W_{ab}} = \mathbb{E}_{\boldsymbol{x}, \boldsymbol{x}_0, t}\left\{ \lambda_t \left[ w_{0a} + \sum_{j=1}^{F} W_{aj}\phi_j(\boldsymbol{x}, t) - \tilde{s}_a(\boldsymbol{x}, t; \boldsymbol{x}_0) \right] \phi_b(\boldsymbol{x}, t) \right\} . \qquad (63)$$

Setting these equal to zero, we have

$$\mathbb{E}_{\boldsymbol{x}, \boldsymbol{x}_0, t}\left\{\lambda_t\right\} w_{0a} + \sum_{j=1}^{F} W_{aj}\mathbb{E}_{\boldsymbol{x}, \boldsymbol{x}_0, t}\left\{\lambda_t\phi_j(\boldsymbol{x}, t)\right\} = \mathbb{E}_{\boldsymbol{x}, \boldsymbol{x}_0, t}\left\{\lambda_t\tilde{s}_a(\boldsymbol{x}, t; \boldsymbol{x}_0)\right\}$$

$$\mathbb{E}_{\boldsymbol{x}, \boldsymbol{x}_0, t}\left\{\lambda_t\phi_b(\boldsymbol{x}, t)\right\} w_{0a} + \sum_{j=1}^{F} W_{aj}\mathbb{E}_{\boldsymbol{x}, \boldsymbol{x}_0, t}\left\{\lambda_t\phi_j(\boldsymbol{x}, t)\phi_b(\boldsymbol{x}, t)\right\} = \mathbb{E}_{\boldsymbol{x}, \boldsymbol{x}_0, t}\left\{\lambda_t\tilde{s}_a(\boldsymbol{x}, t; \boldsymbol{x}_0)\phi_b(\boldsymbol{x}, t)\right\} .$$

The first row tells us that

$$w_{0a} = \frac{1}{\mathbb{E}_t[\lambda_t]}\mathbb{E}_{\boldsymbol{x}, \boldsymbol{x}_0, t}\left\{\lambda_t\tilde{s}_a(\boldsymbol{x}, t; \boldsymbol{x}_0)\right\} - \frac{1}{\mathbb{E}_t[\lambda_t]}\sum_{j=1}^{F} W_{aj}\mathbb{E}_{\boldsymbol{x}, \boldsymbol{x}_0, t}\left\{\lambda_t\phi_j(\boldsymbol{x}, t)\right\} , \qquad (64)$$

or equivalently that the optimal bias term satisfies $\boldsymbol{w}_0^* = \langle\tilde{\boldsymbol{s}}\rangle - \boldsymbol{W}^*\langle\boldsymbol{\phi}\rangle$, where we have used $\langle\cdots\rangle$ to denote averages with respect to $\lambda_t p(\boldsymbol{x}, \boldsymbol{x}_0, t)/\mathbb{E}[\lambda_t]$, and where we have defined the vectors

$$\langle\tilde{\boldsymbol{s}}\rangle := \frac{\mathbb{E}_{\boldsymbol{x}, \boldsymbol{x}_0, t}[\lambda_t\tilde{\boldsymbol{s}}(\boldsymbol{x}, t; \boldsymbol{x}_0)]}{\mathbb{E}_t[\lambda_t]} = \frac{1}{\mathbb{E}_t[\lambda_t]}\int \lambda_t \, \tilde{\boldsymbol{s}}(\boldsymbol{x}, t; \boldsymbol{x}_0) \, p(\boldsymbol{x}, \boldsymbol{x}_0, t) \, d\boldsymbol{x} d\boldsymbol{x}_0 dt$$

$$\langle\boldsymbol{\phi}\rangle := \frac{\mathbb{E}_{\boldsymbol{x}, t}[\lambda_t\boldsymbol{\phi}(\boldsymbol{x}, t)]}{\mathbb{E}_t[\lambda_t]} = \frac{1}{\mathbb{E}_t[\lambda_t]}\int \lambda_t \, \boldsymbol{\phi}(\boldsymbol{x}, t) \, p(\boldsymbol{x}, t) \, d\boldsymbol{x} dt . \qquad (65)$$

Using the first row result, the second row can be written as

$$\langle \phi_b \rangle \left[ \langle \tilde{s}_a \rangle - \sum_{j=1}^F W_{aj} \langle \phi_j \rangle \right] + \sum_{j=1}^F W_{aj} \frac{\mathbb{E}_{\boldsymbol{x},\boldsymbol{x}_0,t} \{ \lambda_t \phi_j(\boldsymbol{x},t) \phi_b(\boldsymbol{x},t) \}}{\mathbb{E}_t[\lambda_t]} = \frac{\mathbb{E}_{\boldsymbol{x},\boldsymbol{x}_0,t} \{ \lambda_t \tilde{s}_a(\boldsymbol{x},t;\boldsymbol{x}_0) \phi_b(\boldsymbol{x},t) \}}{\mathbb{E}_t[\lambda_t]}$$

and hence the second row can be written in terms of matrices

$$\boldsymbol{\Sigma}_{\boldsymbol{\phi}} := \frac{\mathbb{E}_{\boldsymbol{x},t} \{ \lambda_t \left[ \boldsymbol{\phi}(\boldsymbol{x},t) - \langle \boldsymbol{\phi} \rangle \right] \left[ \boldsymbol{\phi}(\boldsymbol{x},t) - \langle \boldsymbol{\phi} \rangle \right]^T \}}{\mathbb{E}_t[\lambda_t]}$$

$$= \frac{1}{\mathbb{E}_t[\lambda_t]} \int \lambda_t \left[ \boldsymbol{\phi}(\boldsymbol{x},t) - \langle \boldsymbol{\phi} \rangle \right] \left[ \boldsymbol{\phi}(\boldsymbol{x},t) - \langle \boldsymbol{\phi} \rangle \right]^T p(\boldsymbol{x},t) \, d\boldsymbol{x}dt$$

$$\boldsymbol{J} := - \frac{\mathbb{E}_{\boldsymbol{x},\boldsymbol{x}_0,t} \{ \lambda_t \left[ \boldsymbol{\phi}(\boldsymbol{x},t) - \langle \boldsymbol{\phi} \rangle \right] \left[ \tilde{\boldsymbol{s}}(\boldsymbol{x},t;\boldsymbol{x}_0) - \langle \tilde{\boldsymbol{s}} \rangle \right]^T \}}{\mathbb{E}_t[\lambda_t]} \tag{66}$$

$$= - \frac{1}{\mathbb{E}_t[\lambda_t]} \int \lambda_t \left[ \boldsymbol{\phi}(\boldsymbol{x},t) - \langle \boldsymbol{\phi} \rangle \right] \left[ \tilde{\boldsymbol{s}}(\boldsymbol{x},t;\boldsymbol{x}_0) - \langle \tilde{\boldsymbol{s}} \rangle \right]^T p(\boldsymbol{x},\boldsymbol{x}_0,t) \, d\boldsymbol{x}d\boldsymbol{x}_0dt .$$

In particular,

$$\boldsymbol{W}^* \boldsymbol{\Sigma}_{\boldsymbol{\phi}} = -\boldsymbol{J}^T \implies \boldsymbol{W}^* = -\boldsymbol{J}^T \boldsymbol{\Sigma}_{\boldsymbol{\phi}}^{-1} , \tag{67}$$

where we have assumed that $\boldsymbol{\Sigma}_{\boldsymbol{\phi}}$ is invertible. This ought to be true, since the feature maps are independent and $p(\boldsymbol{x}|t)$ is a smooth distribution supported on all of $\mathbb{R}^D$ (especially since we are technically only considering $t$ as small as $\epsilon$, the nonzero lower bound, for regularization purposes).

The optimal score is

$$\hat{\boldsymbol{s}}_*(\boldsymbol{x},t) = \boldsymbol{w}_0^* + \boldsymbol{W}^* \boldsymbol{\phi}(\boldsymbol{x},t) = \boldsymbol{J}^T \boldsymbol{\Sigma}_{\boldsymbol{\phi}}^{-1} \left[ \langle \boldsymbol{\phi} \rangle - \boldsymbol{\phi}(\boldsymbol{x},t) \right] + \langle \tilde{\boldsymbol{s}} \rangle . \tag{68}$$

As a side comment, omitting the bias term just removes the mean corrections from the definitions of $\boldsymbol{J}$ and $\boldsymbol{\Sigma}_{\boldsymbol{\phi}}$, as well as the $\langle \boldsymbol{\phi} \rangle$ and $\langle \tilde{\boldsymbol{s}} \rangle$ offsets. Without it, the optimal score is $\hat{\boldsymbol{s}}_*(\boldsymbol{x},t) = \boldsymbol{W}^* \boldsymbol{\phi}(\boldsymbol{x},t) = -\boldsymbol{J}^T \boldsymbol{\Sigma}_{\boldsymbol{\phi}}^{-1} \boldsymbol{\phi}(\boldsymbol{x},t)$, where $\boldsymbol{J}$ and $\boldsymbol{\Sigma}_{\boldsymbol{\phi}}$ are instead defined to be

$$\boldsymbol{\Sigma}_{\boldsymbol{\phi}} := \frac{\mathbb{E}_{\boldsymbol{x},t} \{ \lambda_t \, \boldsymbol{\phi}(\boldsymbol{x},t) \boldsymbol{\phi}(\boldsymbol{x},t)^T \}}{\mathbb{E}_t[\lambda_t]}$$

$$\boldsymbol{J} := - \frac{\mathbb{E}_{\boldsymbol{x},\boldsymbol{x}_0,t} \{ \lambda_t \, \boldsymbol{\phi}(\boldsymbol{x},t) \tilde{\boldsymbol{s}}(\boldsymbol{x},t;\boldsymbol{x}_0)^T \}}{\mathbb{E}_t[\lambda_t]} . \tag{69}$$

In the rest of this appendix, we will assume that the bias term is present.

### F.3 OPTIMUM OF DSM OBJECTIVE GIVEN A FINITE NUMBER OF SAMPLES

Assume we have access to $P \gg 1$ samples $\boldsymbol{x}^{(n)}, \boldsymbol{x}_0^{(n)}, t^{(n)} \sim p(\boldsymbol{x},\boldsymbol{x}_0,t)$, and that we estimate the parameters of the linear score model using naive sample mean estimators

$$\bar{\lambda}_t := \frac{1}{P} \sum_n \lambda^{(n)}$$

$$\hat{\boldsymbol{b}} := \frac{1}{\bar{\lambda}_t} \frac{1}{P} \sum_n \lambda^{(n)} \tilde{\boldsymbol{s}}(\boldsymbol{x}^{(n)}, t^{(n)}; \boldsymbol{x}_0^{(n)})$$

$$\hat{\boldsymbol{\mu}}_{\boldsymbol{\phi}} := \frac{1}{\bar{\lambda}_t} \frac{1}{P} \sum_n \lambda^{(n)} \boldsymbol{\phi}(\boldsymbol{x}^{(n)}, t^{(n)}) \tag{70}$$

$$\hat{\boldsymbol{\Sigma}}_{\boldsymbol{\phi}} := \frac{1}{\bar{\lambda}_t} \frac{1}{P} \sum_n \lambda^{(n)} \left[ \boldsymbol{\phi}(\boldsymbol{x}^{(n)}, t^{(n)}) - \hat{\boldsymbol{\mu}}_{\boldsymbol{\phi}} \right] \left[ \boldsymbol{\phi}(\boldsymbol{x}^{(n)}, t^{(n)}) - \hat{\boldsymbol{\mu}}_{\boldsymbol{\phi}} \right]^T$$

$$\hat{\boldsymbol{J}} := - \frac{1}{\bar{\lambda}_t} \frac{1}{P} \sum_n \lambda^{(n)} \left[ \boldsymbol{\phi}(\boldsymbol{x}^{(n)}, t^{(n)}) - \hat{\boldsymbol{\mu}}_{\boldsymbol{\phi}} \right] \left[ \tilde{\boldsymbol{s}}(\boldsymbol{x}^{(n)}, t^{(n)}; \boldsymbol{x}_0^{(n)}) - \hat{\boldsymbol{b}} \right]^T$$

where we have used $\lambda^{(n)}$ as a slightly less cumbersome shorthand for $\lambda_{t^{(n)}}$. We will not worry about using Bessel's correction in the covariance estimators, and we will see below that $\hat{\boldsymbol{s}}$ is actually unbiased for finite $P$ even if the covariance estimators are not. Our learned score estimator is then

$$\hat{\boldsymbol{s}}_{\boldsymbol{\theta}}(\boldsymbol{x},t) = \hat{\boldsymbol{J}}^T \hat{\boldsymbol{\Sigma}}_{\boldsymbol{\phi}}^{-1} \left[ \hat{\boldsymbol{\mu}}_{\boldsymbol{\phi}} - \boldsymbol{\phi}(\boldsymbol{x},t) \right] + \hat{\boldsymbol{b}} . \tag{71}$$

Note that if the number of samples $P$ is less than $F$, $\hat{\boldsymbol{\Sigma}}_\phi$ is not invertible; in this case, one can either use the Moore-Penrose pseudoinverse, or explicitly include a weight regularization term in the objective, i.e., add to $J_1$ a term of the form

$$J_{reg} := \frac{\xi}{2}\left[\sum_i w_{0i}^2 + \sum_{i,j} W_{ij}^2\right] \tag{72}$$

where the parameter $\xi \geq 0$ controls the importance of this term. Including this term changes the score estimator (Eq. 71) by modifying the inverse that appears:

$$\hat{\boldsymbol{\Sigma}}_\phi^{-1} \to \left[\hat{\boldsymbol{\Sigma}}_\phi + \xi \boldsymbol{I}_F\right]^{-1} . \tag{73}$$

In what follows, if one wants results in the case that such a regularization term is present, note that this replacement of the inverse of the empirical covariance matrix is the only necessary change.

### F.4 LINEAR SCORE MODEL ESTIMATOR IS UNBIASED

We are primarily interested in variance due to $\boldsymbol{x}_0$ (for reasons that will become clear), so we will consider an ensemble of systems for which the $\boldsymbol{x}^{(n)}$ and $t^{(n)}$ sample draws are the same, but the $\boldsymbol{x}_0^{(n)}$ draws are different. Our estimator depends linearly on $\tilde{\boldsymbol{s}}$, the quantity through which it depends on the $\boldsymbol{x}_0$ samples. In particular,

$$\hat{\boldsymbol{J}}^T \hat{\boldsymbol{\Sigma}}_\phi^{-1}[\hat{\boldsymbol{\mu}}_\phi - \boldsymbol{\phi}(\boldsymbol{x},t)] = \frac{1}{P}\sum_n \frac{\lambda^{(n)}}{\bar{\lambda}_t}\left[\tilde{\boldsymbol{s}}(\boldsymbol{x}^{(n)},t^{(n)};\boldsymbol{x}_0^{(n)}) - \hat{\boldsymbol{b}}\right]\left[\boldsymbol{\phi}(\boldsymbol{x}^{(n)},t^{(n)}) - \hat{\boldsymbol{\mu}}_\phi\right]^T \hat{\boldsymbol{\Sigma}}_\phi^{-1}[\boldsymbol{\phi}(\boldsymbol{x},t) - \hat{\boldsymbol{\mu}}_\phi]$$

$$= \frac{1}{P}\sum_n \frac{\lambda^{(n)}}{\bar{\lambda}_t}\tilde{\boldsymbol{s}}(\boldsymbol{x}^{(n)},t^{(n)};\boldsymbol{x}_0^{(n)})\left[\boldsymbol{\phi}(\boldsymbol{x}^{(n)},t^{(n)}) - \hat{\boldsymbol{\mu}}_\phi\right]^T \hat{\boldsymbol{\Sigma}}_\phi^{-1}[\boldsymbol{\phi}(\boldsymbol{x},t) - \hat{\boldsymbol{\mu}}_\phi] ,$$

so

$$\hat{\boldsymbol{s}}_{\boldsymbol{\theta}}(\boldsymbol{x},t) = \frac{1}{P}\sum_n \frac{\lambda^{(n)}}{\bar{\lambda}_t}Q(\boldsymbol{x}^{(n)},t^{(n)};\boldsymbol{x},t)\,\tilde{\boldsymbol{s}}(\boldsymbol{x}^{(n)},t^{(n)};\boldsymbol{x}_0^{(n)}) \tag{74}$$

where we have defined the kernel function

$$Q(\boldsymbol{x},t;\boldsymbol{x}',t') := 1 + [\boldsymbol{\phi}(\boldsymbol{x},t) - \hat{\boldsymbol{\mu}}]^T \hat{\boldsymbol{\Sigma}}_\phi^{-1}[\boldsymbol{\phi}(\boldsymbol{x}',t') - \hat{\boldsymbol{\mu}}] . \tag{75}$$

To see that this estimator is unbiased (when the model is sufficiently expressive), suppose the true score has the form of our linear estimator, i.e.,

$$\boldsymbol{s}(\boldsymbol{x},t) = \boldsymbol{w}_0^* + \boldsymbol{W}^*\boldsymbol{\phi}(\boldsymbol{x},t) = \boldsymbol{W}^*[\boldsymbol{\phi}(\boldsymbol{x},t) - \langle\boldsymbol{\phi}\rangle] , \tag{76}$$

where we have used the fact that $\mathbb{E}_{\boldsymbol{x}}[\boldsymbol{s}] = \langle\boldsymbol{s}\rangle = \boldsymbol{0}$. Next, note that

$$\frac{1}{P}\sum_n \frac{\lambda^{(n)}}{\bar{\lambda}_t}Q(\boldsymbol{x}^{(n)},t^{(n)};\boldsymbol{x},t) = 1 . \tag{77}$$

Averaging our estimator over $\boldsymbol{x}_0$ sample draws yields

$$\mathbb{E}[\hat{\boldsymbol{s}}_{\boldsymbol{\theta}}(\boldsymbol{x},t)] = \frac{1}{P}\sum_n \frac{\lambda^{(n)}}{\bar{\lambda}_t}Q(\boldsymbol{x}^{(n)},t^{(n)};\boldsymbol{x},t)\,\boldsymbol{W}^*[\boldsymbol{\phi}(\boldsymbol{x}^{(n)},t^{(n)}) - \hat{\boldsymbol{\mu}} + \hat{\boldsymbol{\mu}} - \langle\boldsymbol{\phi}\rangle]$$

$$= \frac{1}{P}\sum_n \frac{\lambda^{(n)}}{\bar{\lambda}_t}Q(\boldsymbol{x}^{(n)},t^{(n)};\boldsymbol{x},t)\,\boldsymbol{W}^*[\boldsymbol{\phi}(\boldsymbol{x}^{(n)},t^{(n)}) - \hat{\boldsymbol{\mu}}] + \boldsymbol{W}^*(\hat{\boldsymbol{\mu}} - \langle\boldsymbol{\phi}\rangle)$$

$$= \frac{1}{P}\sum_n \frac{\lambda^{(n)}}{\bar{\lambda}_t}\boldsymbol{W}^*[\boldsymbol{\phi}(\boldsymbol{x}^{(n)},t^{(n)}) - \hat{\boldsymbol{\mu}}]\left[\boldsymbol{\phi}(\boldsymbol{x}^{(n)},t^{(n)}) - \hat{\boldsymbol{\mu}}\right]^T \hat{\boldsymbol{\Sigma}}_\phi^{-1}(\boldsymbol{\phi}(\boldsymbol{x},t) - \hat{\boldsymbol{\mu}}) + \boldsymbol{W}^*(\hat{\boldsymbol{\mu}} - \langle\boldsymbol{\phi}\rangle)$$

$$= \boldsymbol{W}^*(\boldsymbol{\phi}(\boldsymbol{x},t) - \hat{\boldsymbol{\mu}}) + \boldsymbol{W}^*(\hat{\boldsymbol{\mu}} - \langle\boldsymbol{\phi}\rangle)$$

$$= \boldsymbol{W}^*(\boldsymbol{\phi}(\boldsymbol{x},t) - \langle\boldsymbol{\phi}\rangle) ,$$

i.e., it is unbiased. What is worth emphasizing is that this is *exactly* true, and does not require taking any kind of large $P$ limit. In other words, as long as $P$ is large enough that $\hat{\boldsymbol{\Sigma}}_\phi$ is invertible, one recovers the true weights $\boldsymbol{w}_0^*$ and $\boldsymbol{W}^*$, independent of the $\boldsymbol{x}$ and $t$ sample draws. This is why variance due to $\boldsymbol{x}_0$ sample draws matters, and variance due to the other draws does not, at least for this linear model.

### F.5 COMPUTING THE V-KERNEL OF THE LINEAR SCORE MODEL

Computing the V-kernel amounts to computing the covariance of the score model with respect to $x_0$ sample realizations. In the previous section, we computed the mean of our estimator; the covariance calculation will be fairly similar. Note that

$$
\text{Cov}[\hat{s}(z), \hat{s}(z')] = \frac{1}{P^2} \sum_{n,m} \frac{\lambda^{(n)}}{\bar{\lambda}_t} \frac{\lambda^{(m)}}{\bar{\lambda}_t} Q(z^{(n)}; z) Q(z^{(m)}; z') \, \text{Cov}[\tilde{s}(z^{(n)}; x_0^{(n)}), \tilde{s}(z^{(m)}; x_0^{(m)})]
$$

$$
= \frac{1}{P^2} \sum_n \left( \frac{\lambda^{(n)}}{\bar{\lambda}_t} \right)^2 Q(z^{(n)}; z) Q(z^{(n)}; z') \, \text{Cov}[\tilde{s}(z^{(n)}; x_0^{(n)})] \, ,
$$

where we have used $z$ as shorthand for $\{x, t\}$, and the fact that $x_0$ sample draws are independent of one another. Now we will invoke the central limit theorem. Using $C(z) := \text{Cov}[\tilde{s}(z; x_0)]$ as shorthand, when $P$ is very large, to leading order in $1/P$ we have

$$
\text{Cov}[\hat{s}(z), \hat{s}(z')] \approx \frac{1}{P} \int \left( \frac{\lambda_t}{\bar{\lambda}_t} \right)^2 Q(z''; z) Q(z''; z') \, C(z'') \, p(z'') \, dz''
$$

$$
\approx \frac{1}{P} \int \left( \frac{\lambda_t}{\mathbb{E}[\lambda_t]} \right)^2 Q(z''; z) Q(z''; z') \, C(z'') \, p(z'') \, dz''
$$

(78)

where we replace the estimates $\hat{\mu}$ and $\hat{\Sigma}_\phi$ that appear in the kernel function with the true quantities, i.e., we redefine $Q$ to be

$$
Q(x'', t''; x, t) := 1 + [\phi(x'', t'') - \langle \phi \rangle]^T \Sigma_\phi^{-1} [\phi(x, t) - \langle \phi \rangle] \, .
$$

(79)

If the number of features $F$ does *not* scale with the number of samples $P$, then we are done: in the $P \to \infty$ limit, the score estimator covariance, and hence the V-kernel, approach zero. Alternatively, if the number of features $F$ *does* scale with the number of samples $P$, a nontrivial result is possible.

The second term of $Q$, a quadratic form involving the model's feature maps, is the only place in Eq. 78 one can get nontrivial scaling with $F$. Motivated by this observation, define the feature kernel

$$
k(z; z') := \frac{1}{\sqrt{F}} [\phi(z) - \langle \phi \rangle]^T \Sigma_\phi^{-1} [\phi(z') - \langle \phi \rangle] \, .
$$

(80)

Provided that the limit exists and is finite, in the $P \to \infty$ limit (where $F$ may scale with $P$), the asymptotic V-kernel is then

$$
V(z; z') = \lim_{P \to \infty} \frac{F}{P} D_t \, \mathbb{E}_{z''} \left\{ \frac{\lambda_{t''}^2}{\mathbb{E}_t[\lambda_t]^2} k(z; z'') C(z'') k(z''; z') \right\} D_{t'} \, .
$$

(81)

Note also that, in the large $P$ limit, the V-kernel *also* does not depend on the $x$ and $t$ sample draws.

## G    NEURAL NETWORK SCORE ESTIMATOR IN NTK REGIME: DETAILS

In this appendix, we prove Prop. 5.2, which means computing the V-kernel of a fully-connected, infinite-width neural network in the 'lazy' learning (Chizat et al., 2019) regime. Although we focus on an extremely specific type of network here, note that our argument can be straightforwardly adapted to compute the V-kernel of other architectures with NTK limits, like convolutional neural networks (Arora et al., 2019).

### G.1    DEFINITION OF NEURAL NETWORK MODEL

Consider a neural network score function approximator $\hat{\boldsymbol{s}}_{\boldsymbol{\theta}}(\boldsymbol{x}, t)$ trained on the DSM objective (Eq. 4). As elsewhere, we may use $\boldsymbol{z}$ as shorthand for $\{\boldsymbol{x}, t\}$. For concreteness, assume that the network is fully-connected, has $L \geq 1$ layers and $N_*$ trainable parameters, and that each hidden layer has $N$ neurons and an identical pointwise nonlinearity $G$:

$$
\begin{aligned}
a_i^{(0)}(\boldsymbol{z}) &:= \psi_i(\boldsymbol{z}) \\
a_i^{(\ell+1)}(\boldsymbol{z}) &:= G\left(\frac{1}{\sqrt{N}} \sum_j W_{ij}^{(\ell+1)} a_j^{(\ell)}(\boldsymbol{z})\right) \quad \ell = 0, ..., L-2 \\
a_i^{(L)}(\boldsymbol{z}) &:= \frac{1}{\sqrt{N}} \sum_j W_{ij}^{(L)} a_j^{(L-1)}(\boldsymbol{z}) \qquad\qquad \hat{\boldsymbol{s}}_{\boldsymbol{\theta}}(\boldsymbol{z}) := \boldsymbol{a}^{(L)}(\boldsymbol{z}) \, .
\end{aligned}
\tag{82}
$$

The (non-trainable) initial feature maps $\boldsymbol{\psi} := (\psi_1, ..., \psi_{N_0})^T$ account for various preconditioning-related choices. For example, in practice, diffusion models receive time/noise as input only through some time/noise embedding (Ho et al., 2020; Song et al., 2021; Karras et al., 2022).

Although characterizing the gradient descent dynamics of $\hat{\boldsymbol{s}}$ may be difficult in general, if the initial network weights are sampled i.i.d. from a standard normal (i.e., $W_{ij}^{(\ell)} \sim \mathcal{N}(0, 1)$ for all $i$, $j$, and $\ell$), as $N$ is taken to infinity the network output becomes independent of the precise values of the initial weights. Moreover, the network's output throughout training can be written in terms of a kernel function—the so-called NTK—defined by

$$
K^{cc'}(\boldsymbol{z}, \boldsymbol{z}') := \sum_i \mathbb{E}_{\boldsymbol{\theta}} \left\{ \frac{\partial \hat{s}_c(\boldsymbol{z})}{\partial \theta_i} \frac{\partial \hat{s}_{c'}(\boldsymbol{z}')}{\partial \theta_i} \right\}
\tag{83}
$$

where $c$ and $c'$ index different network outputs. In the infinite-width ($N \to \infty$) limit, $K^{cc'}(\boldsymbol{z}, \boldsymbol{z}') = \delta_{cc'} K(\boldsymbol{z}, \boldsymbol{z}')$, i.e., the off-diagonal kernels are identically zero and all kernels along the diagonal are the same (Shan & Bordelon, 2022).

### G.2    LEARNED SCORE AFTER FULL-BATCH GRADIENT DESCENT

**Computing the learned score.**    For simplicity, we assume that our neural network model is trained via full-batch gradient descent on $P$ samples from $p(\boldsymbol{x}, \boldsymbol{x}_0, t)$. Although this assumption does not reflect standard practice (Song et al., 2021; Karras et al., 2022), it makes our computation substantially easier. If we let the dimensionless parameter $\tau$ denote training time, the output evolves via

$$
\frac{d}{d\tau}\hat{\boldsymbol{s}}(\boldsymbol{x}', t') = \mathbb{E}_{\boldsymbol{x}, t, \boldsymbol{x}_0}\left\{ \frac{\lambda_t}{\mathbb{E}_t[\lambda_t]} \frac{\partial \hat{\boldsymbol{s}}(\boldsymbol{x}', t')}{\partial \boldsymbol{\theta}} \frac{\partial \hat{\boldsymbol{s}}(\boldsymbol{x}, t)^T}{\partial \boldsymbol{\theta}} \left[\tilde{\boldsymbol{s}}(\boldsymbol{x}, t; \boldsymbol{x}_0) - \hat{\boldsymbol{s}}(\boldsymbol{x}, t)\right] \right\} \, .
\tag{84}
$$

In the infinite-width limit, we can replace the outer product that appears with the NTK:

$$
\frac{d}{d\tau}\hat{\boldsymbol{s}}(\boldsymbol{x}', t') = \mathbb{E}_{\boldsymbol{x}, t, \boldsymbol{x}_0}\left\{ \frac{\lambda_t}{\mathbb{E}_t[\lambda_t]} K(\boldsymbol{x}', t'; \boldsymbol{x}, t) \left[\tilde{\boldsymbol{s}}(\boldsymbol{x}, t; \boldsymbol{x}_0) - \hat{\boldsymbol{s}}(\boldsymbol{x}, t)\right] \right\} \, .
\tag{85}
$$

Define the Gram matrix $\boldsymbol{K} \in \mathbb{R}^{P \times P}$, the time-weighting matrix $\boldsymbol{\Lambda}_T \in \mathbb{R}^{P \times P}$, the target matrix $\tilde{\boldsymbol{S}} \in \mathbb{R}^{P \times D}$, and the output matrix $\hat{\boldsymbol{S}} \in \mathbb{R}^{P \times D}$ via

$$
\begin{aligned}
K_{ab} &:= K(\boldsymbol{x}^{(a)}, t^{(a)}; \boldsymbol{x}^{(b)}, t^{(b)}) \\
\Lambda_{T,ab} &:= \delta_{ab} \frac{\lambda_{t^{(a)}}}{\mathbb{E}_t[\lambda_t]} \\
\tilde{S}_{ai} &:= \tilde{s}_i(\boldsymbol{x}^{(a)}, t^{(a)}; \boldsymbol{x}_0^{(a)}) \\
\hat{S}_{ai} &:= \hat{s}_i(\boldsymbol{x}^{(a)}, t^{(a)}) .
\end{aligned}
\tag{86}
$$

Eq. 85 implies that

$$
\frac{d}{d\tau} \hat{\boldsymbol{S}} = \frac{1}{P} \boldsymbol{K} \boldsymbol{\Lambda}_T \left( \tilde{\boldsymbol{S}} - \hat{\boldsymbol{S}} \right) .
\tag{87}
$$

Hence, after training, the network's output on the set of samples is given by

$$
\hat{\boldsymbol{S}} = e^{-\boldsymbol{K}\boldsymbol{\Lambda}_T \tau/P} \hat{\boldsymbol{S}}_0 + (\boldsymbol{I} - e^{-\boldsymbol{K}\boldsymbol{\Lambda}_T \tau/P}) \tilde{\boldsymbol{S}}
\tag{88}
$$

where $\tau$ is the total training 'time' and $\hat{\boldsymbol{S}}_0$ is the $P \times D$ matrix containing the network's initial output on the samples. Let $\boldsymbol{k}(\boldsymbol{x}, t)$ denote the $P$-dimensional vector whose $i$-th component is $K(\boldsymbol{x}^{(i)}, t^{(i)}; \boldsymbol{x}, t)$. The network's output given other inputs evolves according to the ODE

$$
\frac{d}{d\tau} \hat{\boldsymbol{s}}(\boldsymbol{x}, t)^T = \frac{1}{P} \boldsymbol{k}(\boldsymbol{x}, t)^T \boldsymbol{\Lambda}_T \left( \tilde{\boldsymbol{S}} - \hat{\boldsymbol{S}} \right) ,
\tag{89}
$$

whose solution is

$$
\hat{\boldsymbol{s}}(\boldsymbol{x}, t)^T = \hat{\boldsymbol{s}}_0(\boldsymbol{x}, t)^T + \boldsymbol{k}(\boldsymbol{x}, t)^T \boldsymbol{K}^{-1} (\boldsymbol{I} - e^{-\boldsymbol{K}\boldsymbol{\Lambda}_T \tau/P}) (\tilde{\boldsymbol{S}} - \hat{\boldsymbol{S}}_0)
\tag{90}
$$

where $\hat{\boldsymbol{s}}_0(\boldsymbol{x}, t)$ is the network's initial output given a $\{\boldsymbol{x}, t\}$ input. If the Gram matrix $\boldsymbol{K}$ is rank-deficient, we must use its Moore-Penrose pseudoinverse. Alternatively, one can avoid this issue by including a weight regularization term in the objective.

**Expressing the learned score in terms of eigenfunctions.** We will find it useful to consider a Mercer decomposition of $K$ with respect to the measure $\lambda_t p(\boldsymbol{x}, t)/\mathbb{E}_t[\lambda_t]$, so that $K$ can be written

$$
K(\boldsymbol{x}, t; \boldsymbol{x}', t') = \sum_k \lambda_k \phi_k(\boldsymbol{x}, t) \phi_k(\boldsymbol{x}', t')
\tag{91}
$$

where the features are orthonormal and complete, i.e.,

$$
\begin{aligned}
\int \frac{\lambda_t}{\mathbb{E}_t[\lambda_t]} \phi_k(\boldsymbol{x}, t) \phi_{k'}(\boldsymbol{x}, t) \, p(\boldsymbol{x}, t) \, d\boldsymbol{x} dt &= \delta_{k,k'} \\
\sum_k \frac{\lambda_t}{\mathbb{E}_t[\lambda_t]} p(\boldsymbol{x}, t) \, \phi_k(\boldsymbol{x}, t) \phi_k(\boldsymbol{x}', t') &= \delta(\boldsymbol{x} - \boldsymbol{x}')\delta(t - t') .
\end{aligned}
\tag{92}
$$

If we assume $\boldsymbol{K}$ has rank $F$ not necessarily equal to P, we can write Eq. 90 in terms of the eigenfunctions associated with the Mercer decomposition by defining the $P \times F$ matrix $\boldsymbol{\Phi}$ with

$$
\Phi_{ak} := \phi_k(\boldsymbol{x}^{(a)}, t^{(a)})
\tag{93}
$$

and noting that $\boldsymbol{K} = \boldsymbol{\Phi} \boldsymbol{\Lambda} \boldsymbol{\Phi}^T$, where $\boldsymbol{\Lambda}$ is the $F \times F$ diagonal matrix of associated eigenvalues. It is useful to observe that

$$
\delta_{kk'} = \mathbb{E}_{\boldsymbol{x},t} \left\{ \frac{\lambda_t}{\mathbb{E}_t[\lambda_t]} \phi_k(\boldsymbol{x}, t) \phi_{k'}(\boldsymbol{x}, t) \right\} = \frac{1}{P} \sum_n \frac{\lambda^{(n)}}{\mathbb{E}_t[\lambda_t]} \phi_k(\boldsymbol{x}^{(n)}, t^{(n)}) \phi_{k'}(\boldsymbol{x}^{(n)}, t^{(n)}) + \mathcal{O}(1/\sqrt{P}) ,
$$

which implies $\boldsymbol{I}_F = \frac{\boldsymbol{\Phi}^T \boldsymbol{\Lambda}_T \boldsymbol{\Phi}}{P}$ to leading order. Similarly, the completeness relation becomes

$$
\boldsymbol{I}_P \approx \frac{\boldsymbol{\Phi} \boldsymbol{\Phi}^T \boldsymbol{\Lambda}_T}{P} = \frac{\boldsymbol{\Lambda}_T \boldsymbol{\Phi} \boldsymbol{\Phi}^T}{P}
\tag{94}
$$

to leading order. Using these identities, we can rewrite Eq. 90 as

$$\hat{s}(x,t)^T = \hat{s}_0(x,t)^T + \left[\phi(x,t)^T \Lambda \Phi^T\right] \left[\frac{\Lambda_T \Phi \Lambda^{-1} \Phi^T \Lambda_T}{P^2}\right] \left[\frac{\Phi}{P}(I - e^{-\Lambda\tau})\Phi^T \Lambda_T\right] (\tilde{S} - \hat{S}_0)$$

$$= \hat{s}_0(x,t)^T + \frac{1}{P}\phi(x,t)^T(I - e^{-\Lambda\tau})\Phi^T \Lambda_T(\tilde{S} - \hat{S}_0) . \tag{95}$$

Equivalently,

$$\hat{s}(x,t) = \hat{s}_0(x,t) + \frac{1}{P}(\tilde{S} - \hat{S}_0)^T \Lambda_T \Phi(I - e^{-\Lambda\tau})\phi(x,t) . \tag{96}$$

Let $S$ denote the $P \times D$ matrix whose entries are the true score evaluated on the set of input samples $\{(x^{(a)}, t^{(a)})\}$. When averaged over $x_0$ sample realizations, our estimator is

$$\mathbb{E}[\hat{s}(x,t)] = \hat{s}_0(x,t) + \frac{1}{P}(S - \hat{S}_0)^T \Lambda_T \Phi(I - e^{-\Lambda\tau})\phi(x,t) \tag{97}$$

which implies

$$\hat{s}(x,t) - \mathbb{E}[\hat{s}(x,t)] = \frac{1}{P}(\tilde{S} - S)^T \Lambda_T \Phi(I - e^{-\Lambda\tau})\phi(x,t) . \tag{98}$$

To make things slightly easier, define the feature kernel[1]

$$k(x,t;x',t') := \frac{1}{\sqrt{F}}\phi(x,t)^T(I - e^{-\Lambda\tau})\phi(x',t') = \frac{1}{\sqrt{F}}\sum_{k=1}^{F}\phi_k(x,t)(1 - e^{-\lambda_k\tau})\phi_k(x',t') . \tag{99}$$

In terms of this kernel, we can write

$$\hat{s}(x,t) - \mathbb{E}[\hat{s}(x,t)] = \frac{\sqrt{F}}{P}\sum_n \frac{\lambda^{(n)}}{\mathbb{E}_t[\lambda_t]}\left[\tilde{s}(x^{(n)}, t^{(n)}; x_0^{(n)}) - s(x^{(n)}, t^{(n)})\right]k(x^{(n)}, t^{(n)}; x, t) . \tag{100}$$

We will use this result in the next subsection to compute the V-kernel of this model.

### G.3   COMPUTING THE V-KERNEL OF THE NTK MODEL

The covariance of the learned score estimator with respect to $x_0$ sample realizations is

$$\text{Cov}[\hat{s}_\theta(z), \hat{s}_\theta(z')] = \frac{F}{P^2}\sum_{n,m}\frac{\lambda^{(n)}}{\mathbb{E}_t[\lambda_t]}\frac{\lambda^{(m)}}{\mathbb{E}_t[\lambda_t]}\text{Cov}\left[\tilde{s}(z^{(n)}; x_0^{(n)}), \tilde{s}(z^{(m)}; x_0^{(m)})\right]k(z^{(n)}; z)k(z^{(m)}; z')$$

$$= \frac{F}{P^2}\sum_n\left(\frac{\lambda^{(n)}}{\mathbb{E}_t[\lambda_t]}\right)^2 \text{Cov}\left[\tilde{s}(z^{(n)}; x_0^{(n)})\right]k(z^{(n)}; z)k(z^{(n)}; z')$$

$$= \frac{F}{P}\int\frac{\lambda_{t''}^2}{\mathbb{E}_t[\lambda_t]^2}C(z'')k(z''; z)k(z''; z')\,p(z'')\,dz'' ,$$

when $P$ is large, where we exploited the independence of the samples in the first step, and the central limit theorem in the second. As elsewhere, we have used $C(z) := \text{Cov}[\tilde{s}(z; x_0)]$ as shorthand.

Finally, the V-kernel is

$$V(z; z') = \lim_{P\to\infty}\frac{F}{P}D_t\,\mathbb{E}_{z''}\left\{\frac{\lambda_{t''}^2}{\mathbb{E}[\lambda_t]^2}k(z; z'')C(z'')k(z''; z')\right\}D_{t'} \tag{101}$$

provided that the limit exists and is finite. Since $F \propto N$, this can happen if $N \propto P$.

Also note that the form of this V-kernel is identical to that of the V-kernel for linear models (c.f. Prop. 5.1), with the only difference being the feature kernel that appears.

---

[1]We have run out of letters, and unfortunately will use $k$ to denote this quantity. Note that it is different from both $K$ and $k$.

The infinite training time limit is of particular interest, since in this limit we expect the model to interpolate all of its (noisy) samples. In this limit, we have

$$
\begin{aligned}
\text{Cov}[\hat{s}_i(\boldsymbol{z}), \hat{s}_j(\boldsymbol{z}')] &= \frac{1}{P} \int \frac{\lambda_{t''}^2}{\mathbb{E}_t[\lambda_t]^2} \, \boldsymbol{\phi}(\boldsymbol{z})^T \boldsymbol{\phi}(\boldsymbol{z}'') \boldsymbol{\phi}(\boldsymbol{z}'')^T \boldsymbol{\phi}(\boldsymbol{z}') C_{ij}(\boldsymbol{z}'') \, p(\boldsymbol{z}'') \, d\boldsymbol{z}'' \\
&= \frac{1}{P} \int \frac{\lambda_{t''}}{\mathbb{E}_t[\lambda_t]} \, \boldsymbol{\phi}(\boldsymbol{z})^T \boldsymbol{\phi}(\boldsymbol{z}'') \left[ \frac{\lambda_{t''}}{\mathbb{E}_t[\lambda_t]} \boldsymbol{\phi}(\boldsymbol{z}'')^T \boldsymbol{\phi}(\boldsymbol{z}') p(\boldsymbol{z}'') \right] C_{ij}(\boldsymbol{z}'') \, d\boldsymbol{z}'' \\
&= \frac{1}{P} \int \frac{\lambda_{t''}}{\mathbb{E}_t[\lambda_t]} \, \boldsymbol{\phi}(\boldsymbol{z})^T \boldsymbol{\phi}(\boldsymbol{z}'') \delta(\boldsymbol{z}'' - \boldsymbol{z}') C_{ij}(\boldsymbol{z}'') \, d\boldsymbol{z}'' \\
&= \frac{1}{P} \frac{\lambda_{t'}}{\mathbb{E}_t[\lambda_t]} \, \boldsymbol{\phi}(\boldsymbol{z})^T \boldsymbol{\phi}(\boldsymbol{z}') C_{ij}(\boldsymbol{z}')
\end{aligned}
\tag{102}
$$

where we have exploited the completeness relation. Now we encounter a subtle technical point. Since $F \neq P$ in general, in the $F, P \to \infty$ limit the quantity

$$
d(\boldsymbol{z}, \boldsymbol{z}') := \frac{1}{P} \frac{\lambda_{t'}}{\mathbb{E}_t[\lambda_t]} \boldsymbol{\Phi}(\boldsymbol{z})^T \boldsymbol{\Phi}(\boldsymbol{z}')
\tag{103}
$$

is not quite equal to the Dirac delta function, but is instead proportional to it. We need to work out the constant of proportionality. To do this, observe that

$$
\sum_n d(\boldsymbol{z}^{(n)}, \boldsymbol{z}^{(n)}) = \frac{1}{P} \sum_n \frac{\lambda^{(n)}}{\mathbb{E}_t[\lambda_t]} \boldsymbol{\Phi}(\boldsymbol{z}^{(n)})^T \boldsymbol{\Phi}(\boldsymbol{z}^{(n)}) \to \sum_{k=1}^F \int \frac{\lambda_t}{\mathbb{E}_t[\lambda_t]} \, \phi_k(\boldsymbol{z}) \phi_k(\boldsymbol{z}) \, p(\boldsymbol{z}) \, d\boldsymbol{z} = F \, .
$$

On the other hand, for the Dirac delta function, we would have

$$
\sum_n \delta(0) = \frac{P}{\Delta \boldsymbol{z}} \, ,
\tag{104}
$$

where $\Delta \boldsymbol{z}$ is some small bin size. This implies

$$
d(\boldsymbol{z}, \boldsymbol{z}') = \frac{F \Delta \boldsymbol{z}}{P} \delta(\boldsymbol{z} - \boldsymbol{z}') \, .
\tag{105}
$$

If we define $\kappa := (F \Delta \boldsymbol{z})/P$, and assume $\kappa$ remains constant as both parameters approach infinity, we finally obtain

$$
\boldsymbol{V}(\boldsymbol{z}; \boldsymbol{z}') = \kappa \boldsymbol{D}_t C(\boldsymbol{z}) \boldsymbol{D}_t \, \delta(\boldsymbol{z} - \boldsymbol{z}') \, .
\tag{106}
$$

## H  RESULTS FOR NOISE PREDICTION FORMULATION

In the main text, we present the problem of training a diffusion model in terms of learning a parameterized score function estimator (see Eq. 4). In practice, one often does not try to directly estimate the score, but the quantity $\boldsymbol{\mathcal{E}}(\boldsymbol{x}, t) := \sigma_t \boldsymbol{s}(\boldsymbol{x}, t)$ (Rombach et al., 2022), since it can be somewhat better-behaved. In this appendix, we note that it is straightforward to port our results to this slightly different setting.

For simplicity, assume that the forward process involves isotropic noise, so that $\boldsymbol{D}_t = \frac{g_t^2}{2} \boldsymbol{I}_D$ and $\boldsymbol{S}_t = \sigma_t^2 \boldsymbol{I}_D$. The DSM objective (Eq. 4) becomes

$$
\begin{aligned}
J_1(\boldsymbol{\theta}) &= \mathbb{E}_{t, \boldsymbol{x}_0, \boldsymbol{x}} \left\{ \frac{\lambda_t}{2\sigma_t^2} \| \sigma_t \hat{\boldsymbol{s}}_{\boldsymbol{\theta}}(\boldsymbol{x}, t) - \sigma_t \tilde{\boldsymbol{s}}(\boldsymbol{x}, t; \boldsymbol{x}_0) \|_2^2 \right\} \\
&= \mathbb{E}_{t, \boldsymbol{x}_0, \boldsymbol{x}} \left\{ \frac{\lambda_t}{2\sigma_t^2} \| \hat{\boldsymbol{\mathcal{E}}}_{\boldsymbol{\theta}}(\boldsymbol{x}, t) - \boldsymbol{\mathcal{E}} \|_2^2 \right\} .
\end{aligned}
\tag{107}
$$

Since $\boldsymbol{\mathcal{E}} := (\alpha_t \boldsymbol{x}_0 - \boldsymbol{x})/\sigma_t$ by definition, $\boldsymbol{\mathcal{E}}$ is normally distributed with mean zero and variance one. The function $\hat{\boldsymbol{\mathcal{E}}}$ can be viewed as taking a noisy sample $\boldsymbol{x}$ as input and outputting the (standardized) noise that was added to the original sample $\boldsymbol{x}_0$.

Importantly, the objective has the same form as before (i.e., a mean-squared error objective comparing an estimator to a target), but the time-weighting function is now $\tilde{\lambda}_t := \lambda_t / \sigma_t^2$.

In this formulation, the PF-ODE reads

$$
\begin{aligned}
\dot{\boldsymbol{x}}_t &= -\beta_t \boldsymbol{x}_t - \frac{g_t^2}{2\sigma_t} \boldsymbol{\mathcal{E}}(\boldsymbol{x}, t) \\
&= -\beta_t \boldsymbol{x}_t - \tilde{D}_t \boldsymbol{\mathcal{E}}(\boldsymbol{x}, t)
\end{aligned}
\tag{108}
$$

where we define

$$
\tilde{D}_t := \frac{g_t^2}{2\sigma_t} .
\tag{109}
$$

Hence, the PF-ODE also has the same form as before. We conclude that our results apply once one makes these identifications, and also makes the slight change

$$
\boldsymbol{C}(\boldsymbol{x}, t) := \sigma_t^2 \mathrm{Cov}_{\boldsymbol{x}_0 | \boldsymbol{x}, t}(\tilde{\boldsymbol{s}})
\tag{110}
$$

since the learning target is now a scaled version $\sigma_t \tilde{\boldsymbol{s}}$ of the proxy score.

# I  RESULTS FOR DENOISER FORMULATION

In the main text, we present the problem of training a diffusion model in terms of learning a parameterized score function estimator (see Eq. 4). An alternative approach formulates the problem in terms of learning a 'denoiser' function, which takes a noise-corrupted sample $\boldsymbol{x}$ as input and outputs a noise-free sample $\boldsymbol{x}_0$. These formulations are mathematically equivalent, but their implementations slightly differ in practice; in this appendix, we note that it is straightforward to port our results to this slightly different setting.

Define the 'optimal' denoiser[2] $\boldsymbol{D}$ in terms of the true score $\boldsymbol{s}$ via

$$\boldsymbol{D}(\boldsymbol{x}, t) := \frac{1}{\alpha_t}\left[\boldsymbol{S}_t\, \boldsymbol{s}(\boldsymbol{x}, t) + \boldsymbol{x}\right] \qquad\qquad \boldsymbol{s}(\boldsymbol{x}, t) = \boldsymbol{S}_t^{-1}\left[\alpha_t \boldsymbol{D}(\boldsymbol{x}, t) - \boldsymbol{x}\right] . \qquad (111)$$

Parameterized estimators of the denoiser and score, which we will denote by $\hat{\boldsymbol{D}}$ and $\hat{\boldsymbol{s}}$, are related in the same way. This means that the DSM objective (Eq. 4) becomes

$$
\begin{aligned}
J_1(\boldsymbol{\theta}) &= \mathbb{E}_{t, \boldsymbol{x}_0, \boldsymbol{x}}\left\{\frac{\lambda_t}{2}\|\hat{\boldsymbol{s}}_{\boldsymbol{\theta}}(\boldsymbol{x}, t) - \tilde{\boldsymbol{s}}(\boldsymbol{x}, t; \boldsymbol{x}_0)\|_2^2\right\} \\
&= \mathbb{E}_{t, \boldsymbol{x}_0, \boldsymbol{x}}\left\{\frac{\lambda_t}{2}\|\boldsymbol{S}_t^{-1}[\alpha_t \hat{\boldsymbol{D}}_{\boldsymbol{\theta}}(\boldsymbol{x}, t) - \boldsymbol{x}] - \boldsymbol{S}_t^{-1}[\alpha_t \boldsymbol{x}_0 - \boldsymbol{x}]\|_2^2\right\} \\
&= \mathbb{E}_{t, \boldsymbol{x}_0, \boldsymbol{x}}\left\{\frac{\lambda_t \alpha_t^2}{2}[\hat{\boldsymbol{D}}_{\boldsymbol{\theta}}(\boldsymbol{x}, t) - \boldsymbol{x}_0]^T \boldsymbol{S}_t^{-2}[\hat{\boldsymbol{D}}_{\boldsymbol{\theta}}(\boldsymbol{x}, t) - \boldsymbol{x}_0]\right\} .
\end{aligned}
\qquad (112)
$$

For simplicity, we will assume the diffusion tensor of the forward process is isotropic ($\boldsymbol{D}_t = \frac{g_t^2}{2}\boldsymbol{I}_D$), which implies $\boldsymbol{S}_t = \sigma_t^2 \boldsymbol{I}_D$. Then

$$J_1(\boldsymbol{\theta}) = \mathbb{E}_{t, \boldsymbol{x}_0, \boldsymbol{x}}\left\{\frac{\lambda_t \alpha_t^2}{2\sigma_t^4}\|\hat{\boldsymbol{D}}_{\boldsymbol{\theta}}(\boldsymbol{x}, t) - \boldsymbol{x}_0\|_2^2\right\} = \mathbb{E}_{t, \boldsymbol{x}_0, \boldsymbol{x}}\left\{\frac{\tilde{\lambda}_t}{2}\|\hat{\boldsymbol{D}}_{\boldsymbol{\theta}}(\boldsymbol{x}, t) - \boldsymbol{x}_0\|_2^2\right\} \qquad (113)$$

where $\tilde{\lambda}_t := \lambda_t \alpha_t^2/\sigma_t^4$. Hence, the objective has the same form as before (a mean-squared error objective comparing an estimator to a target, with a particular time-weighting function).

In this formulation, the PF-ODE reads

$$
\begin{aligned}
\dot{\boldsymbol{x}}_t &= -\beta_t \boldsymbol{x}_t - \frac{g_t^2}{2\sigma_t^2}[\alpha_t \boldsymbol{D}(\boldsymbol{x}_t, t) - \boldsymbol{x}] \\
&= -\left(\beta_t - \frac{g_t^2}{2\sigma_t^2}\right)\boldsymbol{x}_t - \frac{g_t^2}{2\sigma_t^2}\alpha_t \boldsymbol{D}(\boldsymbol{x}_t, t) \\
&= -\tilde{\beta}_t \boldsymbol{x}_t - \tilde{D}_t \boldsymbol{D}(\boldsymbol{x}_t, t)
\end{aligned}
\qquad (114)
$$

where we define

$$\tilde{\beta}_t := \beta_t - \frac{g_t^2}{2\sigma_t^2} \qquad\qquad \tilde{D}_t := \frac{g_t^2}{2\sigma_t^2}\alpha_t . \qquad (115)$$

Hence, the PF-ODE also has the same form as before. We conclude that our results apply once one makes these identifications, and also makes the slight change

$$\boldsymbol{C}(\boldsymbol{x}, t) := \mathrm{Cov}_{\boldsymbol{x}_0|\boldsymbol{x}, t}(\boldsymbol{x}_0) \qquad (116)$$

since the learning target is now $\boldsymbol{x}_0$ rather than the proxy score.

---

[2] There is an unfortunate collision of notation here, with $\boldsymbol{D}$ being used to denote both the diffusion tensor and the denoiser, but it should be fairly clear from context which is which.

## J    BENIGN PROPERTIES OF GENERALIZATION THROUGH VARIANCE

A priori, one may worry that generalization through variance can happen in an 'uncontrolled' fashion, and hence produce generalizations of the data distribution that are somehow 'bad' or 'unreasonable'. In this appendix, we collect properties of generalization through variance that help ensure that this does not happen.

Most of these properties relate to the proxy score covariance, which when the data distribution consists of $M$ examples has the form (see Appendix B)

$$C(x, t) := \mathrm{Cov}_{x_0|x,t}(\tilde{s}) = \alpha_t^2 S_t^{-1} \mathrm{Cov}_{\mathcal{M}}(\mu) S_t^{-1} \tag{117}$$

where

$$p(x_0 = \mu_m | x, t) = \frac{\mathcal{N}(x; \alpha_t \mu_m, S_t)}{\sum_{m'} \mathcal{N}(x; \alpha_t \mu_{m'}, S_t)} . \tag{118}$$

For the sake of this appendix, we will focus mainly on the case where the forward process is isotropic, which means $G_t = g_t I_D$, $D_t = \frac{g_t^2}{2} I_D$, and $S_t = \sigma_t^2 I_D$. In this case,

$$C(x, t) = \frac{\alpha_t^2}{\sigma_t^4} \mathrm{Cov}_{x_0|x,t}(x_0)$$

$$p(x_0 = \mu_m | x, t) = \frac{\mathcal{N}(x; \alpha_t \mu_m, \sigma_t^2 I_D)}{\sum_{m'} \mathcal{N}(x; \alpha_t \mu_{m'}, \sigma_t^2 I_D)} = \mathrm{softmax}\left( -\frac{\|x - \alpha_t \mu_m\|_2^2}{2\sigma_t^2} \right) . \tag{119}$$

### J.1    SINGLE POINTS ARE NOT GENERALIZED

Suppose that the data distribution consists of a single point at $x_0 = \mu$, so that $M = 1$. How do diffusion models generalize a single point? Since

$$p(x_0 = \mu | x, t) = \frac{\mathcal{N}(x; \alpha_t \mu, \sigma_t^2 I_D)}{\mathcal{N}(x; \alpha_t \mu, \sigma_t^2 I_D)} = 1 , \tag{120}$$

the covariance of $x_0$ is zero. This makes sense, since given a pair $(x, t)$, there is no uncertainty about the training example $x_0$ that generated it.

This implies that the covariance of $x_0$ given $x$ and $t$ is zero, and hence (if the forward process is isotropic) that the proxy score covariance is zero. For the naive score estimator and NTK-regime neural network, this implies that the V-kernel is always zero, and hence that generalization through variance does not occur. This is reasonable, among other reasons because there is no reason to introduce anisotropy if it is not present in the data.

### J.2    DIMENSIONALITY OF DATA DISTRIBUTION IS PRESERVED

Suppose that the data distribution lies entirely within a subspace of dimension $r < D$, so that the components of each $\mu_m$ along the non-subspace dimensions are equal (to $\mu'_k$ along dimension $k$, say). It seems intuitively reasonable not to substantially modify probability mass outside of this subspace. We clearly have

$$\mathbb{E}_{x_0|x,t}[x_{0k}] = \mu'_k \tag{121}$$

for all non-subspace directions $k$. This has the following consequence. Let $\ell$ denote some other (possibly within the subspace, possibly not) direction of state space. The $k$-$\ell$ covariance is

$$\mathrm{Cov}_{x_0|x,t}[(x_{0k} - \mu'_k)(x_{0\ell} - \mathbb{E}_{x_0|x,t}[x_{0\ell}])] = 0 \tag{122}$$

since $x_{0k}$ is always equal to $\mu'_k$. Hence, $C_{ij}$ (and by extension, $V_{ij}$) is only nonzero if $i$ and $j$ are directions along which the data distribution varies, which means generalization through variance only happens along the 'data manifold'.

### J.3    VARIANCE IS NOT ADDED FAR FROM DATA DISTRIBUTION EXAMPLES

Since $p(x_0|x, t)$ has the form of a softmax function, taking $x$ extremely large is analogous to taking the temperature parameter of a typical softmax to be extremely small. For $x$ far from the support

of the data distribution, to good approximation $p(\boldsymbol{x} = \boldsymbol{\mu}_m | \boldsymbol{x}, t) = \delta_{m, m_C(\boldsymbol{x}, t)}$, where $m_C(\boldsymbol{x}, t)$ denotes the index of the data point closest (in terms of Euclidean distance) to $\boldsymbol{x}$. Since $p(\boldsymbol{x}_0 | \boldsymbol{x}, t)$ collapses to a Kronecker delta function, its covariance goes to zero, and hence the V-kernel also goes to zero.

## J.4 FOLLOWING AVERAGE SCORE FIELD IS MOST LIKELY

A straightforward consequence of the effective reverse process (Eq. 8) is that

$$\frac{d}{dt} \langle \boldsymbol{x} \rangle = \langle f(\boldsymbol{x}, t) \rangle \tag{123}$$

where the angular brackets here denote averages with respect to $[q(\boldsymbol{x}, t)]$. Hence, although the effective reverse process involves noise, that noise has zero mean, so on average the system follows PF-ODE dynamics that use the ensemble-averaged score estimator. If the estimator is unbiased, this is just the usual PF-ODE. In other words, even in our setting, *on average* PF-ODE dynamics will reproduce training examples.

There is a slight technical subtlety here, which is the fact that $\langle \boldsymbol{f}(\boldsymbol{x}, t) \rangle \neq \boldsymbol{f}(\langle \boldsymbol{x} \rangle, t)$ in general, but this distinction becomes unimportant for very small noise scales, which most strongly influence whether memorization occurs.

## J.5 TRAINING DATA ARE MORE LIKELY TO BE SAMPLED WHEN NOISE IS SMALL

Similar to the previous point, if the prefactor $\kappa$ that controls the size of the V-kernel is not too large, trajectories that follow PF-ODE dynamics are more *likely* than trajectories that do not. Note that there is a distinction between the *most likely* trajectories of a stochastic process, and the *average* trajectory associated with that process, although here one expects them to be fairly similar.

This idea can be formalized in the context of a semiclassical analysis (see Appendix **??**), which shows that the probability of a given path goes like $\exp(-\mathcal{S}/\kappa)$, where

$$\mathcal{S}[\boldsymbol{x}_t] := \int_\epsilon^T \int_\epsilon^T \frac{1}{2} \left[ \dot{\boldsymbol{x}}_t - \boldsymbol{f}(\boldsymbol{x}_t, t) \right]^T \boldsymbol{Q}(\boldsymbol{x}_t, t; \boldsymbol{x}_{t'}, t') \left[ \dot{\boldsymbol{x}}_{t'} - \boldsymbol{f}(\boldsymbol{x}_{t'}, t') \right] \, dt dt' \tag{124}$$

for some matrix $\boldsymbol{Q}$ that functions as the inverse of $\boldsymbol{V}$. The point we would like to make here corresponds to the observation that this action attains its smallest value when $\dot{\boldsymbol{x}}_t = \boldsymbol{f}(\boldsymbol{x}_t, t)$ for all times $t$. That is, trajectories which follow PF-ODE dynamics are more likely than those that deviate from it, with deviations being penalized more harshly as $\kappa$ is made smaller.

## K    MEMORIZATION AND THE V-KERNEL IN THE SMALL NOISE LIMIT

It's clear how the V-kernel affects a given sampling time step, but how does it affect the overall learned distribution? This question is difficult to answer analytically, since in general the effective reverse process is not exactly solvable. Even without the difficulties related to state-dependent noise and nontrivial temporal autocorrelations that the V-kernel introduces, PF-ODE dynamics are only exactly solvable in special circumstances, e.g., when the data distribution is Gaussian (Wang & Vastola, 2023; 2024).

It is slightly easier to relate the V-kernel to the (average) learned distribution $[q(\boldsymbol{x}_0)]$ when the overall magnitude of the V-kernel is small, since in this limit one can invoke a semiclassical approximation (Kleinert, 2006), which is the path integral analogue of a saddlepoint approximation. This limit plausibly applies if the prefactor $\kappa$ (which is proportional to the ratio $F/P$ for the latter two kinds of models we consider) is small, or equivalently if the model is somewhat underparameterized.

A proper treatment of this topic involves the careful manipulation of notoriously tricky mathematical objects like functional determinants; to sidestep these issues, and be somewhat more brief, we will proceed in a somewhat heuristic fashion here. If one of the following steps appears unclear, we advise the reader to examine it in discrete rather than continuous time, where the associated issues are less severe.

### K.1    SETTING UP THE SEMICLASSICAL APPROXIMATION

First, define the prefactor-divided V-kernel $\tilde{\boldsymbol{V}}$ as

$$\tilde{\boldsymbol{V}}(\boldsymbol{z};\boldsymbol{z}') := \boldsymbol{V}(\boldsymbol{z};\boldsymbol{z}')/\kappa \,, \tag{125}$$

and define the 'inverse' $\boldsymbol{Q}$ of the V-kernel as the vector-valued matrix satisfying

$$\int_{\epsilon}^{T} \boldsymbol{Q}(\boldsymbol{x}_t,t;\boldsymbol{x}_{t''},t'')\tilde{\boldsymbol{V}}(\boldsymbol{x}_{t''},t'';\boldsymbol{x}_{t'},t')\,dt'' = \int_{\epsilon}^{T} \tilde{\boldsymbol{V}}(\boldsymbol{x}_t,t;\boldsymbol{x}_{t''},t'')\boldsymbol{Q}(\boldsymbol{x}_{t''},t'';\boldsymbol{x}_{t'},t')\,dt'' = \boldsymbol{I}_D\,\delta(t-t')\,.$$

Our path integral expression for $[q(\boldsymbol{x}_0)]$ has the form (see Appendix D)

$$[q(\boldsymbol{x}_0)] = \int \mathcal{D}[\boldsymbol{x}_t]\mathcal{D}[\boldsymbol{p}_t]\,\exp\left\{-\mathcal{S}[\boldsymbol{x}_t,\boldsymbol{p}_t]\right\}\mathcal{N}(\boldsymbol{x}_T;\boldsymbol{0},\boldsymbol{S}_T)\,, \tag{126}$$

where the 'action' $\mathcal{S}$ is

$$\mathcal{S}[\boldsymbol{x}_t,\boldsymbol{p}_t] := \int_{\epsilon}^{T} -i\boldsymbol{p}_t^T\left[\dot{\boldsymbol{x}}_t - \boldsymbol{f}(\boldsymbol{x}_t,t)\right]\,dt + \frac{\kappa}{2}\int_{\epsilon}^{T}\int_{\epsilon}^{T}\boldsymbol{p}_t^T\tilde{\boldsymbol{V}}(\boldsymbol{x}_t,t;\boldsymbol{x}_{t'},t')\boldsymbol{p}_{t'}\,dtdt'\,, \tag{127}$$

and where the functional integral is over all paths $\boldsymbol{x}(t)$ which have $\boldsymbol{x}(0) = \boldsymbol{x}_0$ and $\boldsymbol{x}(T) = \boldsymbol{x}_T$, and all possible $\boldsymbol{p}(t)$ paths. The explicit form of $\boldsymbol{f}$ is

$$\boldsymbol{f}(\boldsymbol{x},t) := -\beta_t\boldsymbol{x} - \boldsymbol{D}_t\boldsymbol{s}_{avg}(\boldsymbol{x},t)\,, \tag{128}$$

although for this analysis it is not relevant.

Since the action is quadratic in the 'momenta' variables $\boldsymbol{p}_t$, we can perform the associated Gaussian integrals exactly to obtain a reduced path integral with

$$[q(\boldsymbol{x}_0)] = \int \mathcal{D}[\boldsymbol{x}_t]\mathcal{D}[\boldsymbol{p}_t]\,\exp\left\{-\mathcal{S}[\boldsymbol{x}_t]/\kappa\right\}\,F(\{\boldsymbol{x}_t\})\,\mathcal{N}(\boldsymbol{x}_T;\boldsymbol{0},\boldsymbol{S}_T)$$

$$\mathcal{S}[\boldsymbol{x}_t] = \int_{\epsilon}^{T}\int_{\epsilon}^{T}\left[\dot{\boldsymbol{x}}_t - \boldsymbol{f}(\boldsymbol{x}_t,t)\right]^T\boldsymbol{Q}(\boldsymbol{x}_t,t;\boldsymbol{x}_{t'},t')\left[\dot{\boldsymbol{x}}_{t'} - \boldsymbol{f}(\boldsymbol{x}_{t'},t')\right]\,dtdt' \tag{129}$$

where the factor $F$ involves a functional determinant

$$F(\{\boldsymbol{x}_t\}) := \frac{1}{\sqrt{\det\boldsymbol{V}}} \tag{130}$$

due to the Hessian of the action. Note that $\boldsymbol{Q}$ is positive semidefinite since $\boldsymbol{V}$ is, and hence the minimum possible value of $\mathcal{S}$ is zero.

### K.2 SEMICLASSICAL APPROXIMATION OF THE LEARNED DISTRIBUTION

One can now invoke the semiclassical approximation of this path integral assuming $\kappa$ is sufficiently small. Relevant to this approximation is the 'classical' action $\mathcal{S}_{cl}(\boldsymbol{x}_0, \boldsymbol{x}_T)$, which quantifies the likelihood of $\boldsymbol{x}_t$ following the most likely (i.e., 'classical') path from $\boldsymbol{x}_T$ to $\boldsymbol{x}_0$:

$$\mathcal{S}_{cl}(\boldsymbol{x}_0, \boldsymbol{x}_T) := \min_{\boldsymbol{x}(t):\boldsymbol{x}(0)=\boldsymbol{x}_0,\boldsymbol{x}(T)=\boldsymbol{x}_T} \mathcal{S}[\boldsymbol{x}_t] \ . \tag{131}$$

Also relevant is the Hessian of this quantity evaluated at the least action path, which reads

$$\mathcal{H} = \frac{1}{\kappa} \frac{\delta^2 \mathcal{S}}{\delta \boldsymbol{x}_t \delta \boldsymbol{x}_{t'}} = \frac{1}{\kappa} \left( \frac{d}{dt} - \boldsymbol{J_f} \right)^T \boldsymbol{Q} \left( \frac{d}{dt} - \boldsymbol{J_f} \right) + \text{additional terms} \tag{132}$$

where $\boldsymbol{J_f}$ is the Jacobian of $\boldsymbol{f}$. The additional terms will vanish after we make one more approximation, so we ignore them. The (functional) determinant of this Hessian is

$$\det \mathcal{H}(\boldsymbol{x}_0, \boldsymbol{x}_T) = \det \left[ \frac{1}{\kappa} \left( \frac{d}{dt} - \boldsymbol{J_f} \right)^T \boldsymbol{Q} \left( \frac{d}{dt} - \boldsymbol{J_f} \right) + \text{additional terms} \right] \ . \tag{133}$$

Although our notation here is deliberately somewhat vague to sidestep various technical details, the above quantity depends on the entire path $\{\boldsymbol{x}_t\}$, and the values of the Jacobian and $\boldsymbol{Q}$ along it.

The semiclassical approximation says that

$$[q(\boldsymbol{x}_0)] \approx \int \frac{d\boldsymbol{x}_T}{\sqrt{(2\pi)^D}} \ \exp\left\{ -\mathcal{S}_{cl}(\boldsymbol{x}_0, \boldsymbol{x}_T)/\kappa \right\} \ \frac{\mathcal{N}(\boldsymbol{x}_T; \boldsymbol{0}, \boldsymbol{S}_T)}{\sqrt{\det \mathcal{H}(\boldsymbol{x}_0, \boldsymbol{x}_T) \det \boldsymbol{V}(\boldsymbol{x}_0, \boldsymbol{x}_T)}} \ . \tag{134}$$

We can invoke Laplace's method in order to approximately evaluate the $\boldsymbol{x}_T$ integral, and hence obtain an expression for the (average) learned distribution that does not involve any integrals. The classical action $\mathcal{S}$ can be expanded with respect to $\boldsymbol{x}_T^*(\boldsymbol{x}_0)$, the *most likely* (in terms of minimizing $\mathcal{S}_{cl}$) noise seed $\boldsymbol{x}_T$ given the endpoint $\boldsymbol{x}_0$:

$$\mathcal{S}_{cl}(\boldsymbol{x}_0, \boldsymbol{x}_T) \approx \mathcal{S}_{cl}(\boldsymbol{x}_0, \boldsymbol{x}_T^*(\boldsymbol{x}_0)) + \frac{1}{2} [\boldsymbol{x}_T - \boldsymbol{x}_T^*]^T \frac{\partial^2 \mathcal{S}_{cl}(\boldsymbol{x}_0, \boldsymbol{x}_T^*(\boldsymbol{x}_0))}{\partial \boldsymbol{x}_T \partial \boldsymbol{x}_T} [\boldsymbol{x}_T - \boldsymbol{x}_T^*] \ . \tag{135}$$

But the classical action takes its minimum possible value—i.e., zero—when $\boldsymbol{x}_T$ is chosen to be the unique noise seed that corresponds to deterministic PF-ODE dynamics (i.e., $\dot{\boldsymbol{x}}_t = \boldsymbol{f}(\boldsymbol{x}_t, t)$ at all times $t$), so we just have

$$\mathcal{S}_{cl}(\boldsymbol{x}_0, \boldsymbol{x}_T) \approx \frac{1}{2} [\boldsymbol{x}_T - \boldsymbol{x}_T^*]^T \frac{\partial^2 \mathcal{S}_{cl}(\boldsymbol{x}_0, \boldsymbol{x}_T^*(\boldsymbol{x}_0))}{\partial \boldsymbol{x}_T \partial \boldsymbol{x}_T} [\boldsymbol{x}_T - \boldsymbol{x}_T^*] \ . \tag{136}$$

By Laplace's method, which is also usable due to the smallness of $\kappa$, we obtain

$$[q(\boldsymbol{x}_0)] \approx \frac{1}{\sqrt{\det \left( \frac{1}{\kappa} \frac{\partial^2 \mathcal{S}_{cl}(\boldsymbol{x}_0, \boldsymbol{x}_T^*(\boldsymbol{x}_0))}{\partial \boldsymbol{x}_T \partial \boldsymbol{x}_T} \right)}} \frac{\mathcal{N}(\boldsymbol{x}_T^*(\boldsymbol{x}_0); \boldsymbol{0}, \boldsymbol{S}_T)}{\sqrt{\det \mathcal{H}(\boldsymbol{x}_0, \boldsymbol{x}_T^*(\boldsymbol{x}_0)) \det \boldsymbol{V}(\boldsymbol{x}_0, \boldsymbol{x}_T^*(\boldsymbol{x}_0))}} \ . \tag{137}$$

If we only consider 'classical' paths with $\dot{\boldsymbol{x}}_t = \boldsymbol{f}(\boldsymbol{x}_t, t)$, as in this approximation, the additional terms in Eq. 132 vanish. This means

$$\det \mathcal{H}(\boldsymbol{x}_0, \boldsymbol{x}_T^*(\boldsymbol{x}_0)) = \det \left[ \frac{1}{\kappa} \left( \frac{d}{dt} - \boldsymbol{J_f} \right)^T \boldsymbol{Q} \left( \frac{d}{dt} - \boldsymbol{J_f} \right) \right]$$
$$= \det \left( \frac{\boldsymbol{Q}}{\kappa} \right) \det \left( \frac{d}{dt} - \boldsymbol{J_f} \right)^2 \tag{138}$$

where this manipulation can be more formally justified if one works in discrete time. But since $\boldsymbol{Q}/\kappa$ is the inverse of $\boldsymbol{V}$, and since

$$\det \left( \frac{d}{dt} - \boldsymbol{J_f} \right) \approx \det \left( \prod_{t=1}^{T} [\boldsymbol{I} - \boldsymbol{J}_t \Delta t] \right) \approx \det \exp \left\{ -\int_{\epsilon}^{T} \boldsymbol{J_f}(t) \, dt \right\} \ , \tag{139}$$

where we have used a discrete time argument to compute the determinant, we have

$$
\det \mathcal{H}(\boldsymbol{x}_0, \boldsymbol{x}_T^*(\boldsymbol{x}_0)) \det \boldsymbol{V}(\boldsymbol{x}_0, \boldsymbol{x}_T^*(\boldsymbol{x}_0)) = \det \boldsymbol{V} \det \boldsymbol{V}^{-1} \det \exp\left\{ -2 \int_\epsilon^T \boldsymbol{J_f}(t)\, dt \right\}
$$

$$
= \det \exp\left\{ -2 \int_\epsilon^T \boldsymbol{J_f}(t)\, dt \right\} .
$$

(140)

Note that this determinant can be simplified somewhat:

$$
\log \det \exp\left\{ -2 \int_\epsilon^T \boldsymbol{J_f}(t)\, dt \right\} = -2 \int_\epsilon^T \mathrm{tr}[\boldsymbol{J_f}(t)]\, dt = -2 \int_\epsilon^T \nabla_{\boldsymbol{x}_t} \cdot \boldsymbol{f}(\boldsymbol{x}_t, t)\, dt \tag{141}
$$

where the $\boldsymbol{x}_t$ that appears in the expression above follows deterministic PF-ODE dynamics. Finally,

$$
[q(\boldsymbol{x}_0)] \approx \frac{1}{\sqrt{\det\left( \frac{1}{\kappa} \frac{\partial^2 \mathcal{S}_{cl}(\boldsymbol{x}_0, \boldsymbol{x}_T^*(\boldsymbol{x}_0))}{\partial \boldsymbol{x}_T \partial \boldsymbol{x}_T} \right)}} \ \exp\left\{ \log \mathcal{N}(\boldsymbol{x}_T^*(\boldsymbol{x}_0); \boldsymbol{0}, \boldsymbol{S}_T) + \int_\epsilon^T \nabla_{\boldsymbol{x}_t} \cdot \boldsymbol{f}(\boldsymbol{x}_t, t)\, dt \right\} .
$$

(142)

At this point, we can make a crucial observation: the argument of the exponential is *precisely* the instantaneous change of variables formula that can be used to compute the log-likelihood $p(\boldsymbol{x}_0|\epsilon)$ (Song et al., 2021; Chen et al., 2018). This immediately implies

$$
[q(\boldsymbol{x}_0)] \approx p(\boldsymbol{x}_0|\epsilon)\, \frac{1}{\sqrt{\det\left( \frac{1}{\kappa} \frac{\partial^2 \mathcal{S}_{cl}(\boldsymbol{x}_0, \boldsymbol{x}_T^*(\boldsymbol{x}_0))}{\partial \boldsymbol{x}_T \partial \boldsymbol{x}_T} \right)}} . \tag{143}
$$

We conclude that, at least in the small $\kappa$ regime, the learned distribution is equal to the memorized ($\epsilon$-noise-corrupted) data distribution, times the determinant of a Hessian that quantifies the likelihood of deviating from PF-ODE dynamics. This Hessian depends on the V-kernel, since the classical action depends on its inverse $\boldsymbol{Q}$, so it is precisely here that the V-kernel can influence generalization.

The required Hessian appears difficult to compute in general. Incidentally, since the relevant action (Eq. 129) is generically not local in time, it is also hard to derive a Hamilton-Jacobi-type differential equation satisfied by this Hessian.

## K.3 Quantifying memorization in the semiclassical regime

Suppose we quantify memorization by computing the Kullback-Leibler (KL) divergence between the data distribution $p_{data}$ and the (average) learned distribution $[q]$:

$$
E_{mem} := D_{KL}(p_{data} \| [q]) = \int p_{data}(\boldsymbol{x}_0) \log \frac{p_{data}(\boldsymbol{x}_0)}{[q(\boldsymbol{x}_0)]}\, d\boldsymbol{x}_0 . \tag{144}
$$

By our semiclassical approximation result (Eq. 143), this is just

$$
E_{mem} = D_{KL}(p_{data} \| p_\epsilon) + \frac{1}{2} \int p_{data}(\boldsymbol{x}_0) \log \det \left( \frac{1}{\kappa} \frac{\partial^2 \mathcal{S}_{cl}(\boldsymbol{x}_0, \boldsymbol{x}_T^*(\boldsymbol{x}_0))}{\partial \boldsymbol{x}_T \partial \boldsymbol{x}_T} \right)\, d\boldsymbol{x}_0 , \tag{145}
$$

where $p_\epsilon := p(\boldsymbol{x}_0|\epsilon)$. Hence, the curvature of the classical action near $\boldsymbol{x}_T^*(\boldsymbol{x}_0)$ strongly controls the extent to which the data distribution is memorized, especially when $\epsilon$ is taken to be small, in which case the first term is negligible.

