# OpenReview forum: "Generalization through variance: how noise shapes inductive biases in diffusion models"
_ICLR.cc/2025/Conference — ICLR 2025 Poster_

### Official Review · Reviewer_AyYq · 2024-11-04

**Soundness:** 3
**Presentation:** 3
**Contribution:** 3
**Rating:** 6
**Confidence:** 2

**Summary:**

The authors study the generalization in diffusion models. They highlight six key factors influencing generalization: noisy objective functions, forward process dynamics, nonlinear score dependencies, model capacity, architectural features, and the structure of the training set. They
attribute the generalization of diffusion models to the fact that the training objective is in fact equal to the ground truth score function only in expectation. The key statement is that generalization occurs if and only if the V-kernel is not zero. Then they characterize the generalizability of diffusion models by analyzing the structures of V-kernel under several simplified conditions.

**Strengths:**

1. Expressing PF-ODE in terms of the path integral is novel and offers important insight i.e, generalization occurs if and only if the V-kernel is nonzero.

2. The paper provides comprehensive study on three distinct cases through the V-kenrl: (i) memorization (ii) linear model (iii) NTK model. These results demonstrate V-kernel as a useful tool toward understanding generalization.

**Weaknesses:**

(i) The authors claim that generalization occurs if and only if V-kernel is nonzero since otherwise the reverse sampling process can only produce training examples. However, this definition seems restricted since when diffusion models generalize, they not only differ from the PF-ODE dynamics, but generate "high quality" samples as well. In other words, the V-kernel should have certain benign properties. This point is not adequately addressed in the current paper.

(ii) In section 5 the authors derive the analytical form of the V-kernels under three circumstances. However, the resulting equations (9), (13) and (17) are really hard for me to interpret. Can the authors provide more explanation on the unique properties of these different V-kernels and how they shape the distributions corresponding to the reverse sampling process?

(iii) As a researcher focuses on empirical works, I don't find the results of this paper very useful. I think the paper could benefit from a further discussion on how the results can help improving practical diffusion models.

(iv) Overall, despite the weaknesses I listed above, I think the results are interesting and worth being published.

**Questions:**

See my questions in the weakness part.

---

> ### Author Response · Authors · 2024-11-28
> **Review response**
>
> We thank the reviewer for their time and thoughtful comments. See below for a point-by-point response.
>
> > (i) The authors claim that generalization occurs if and only if V-kernel is nonzero since otherwise the reverse sampling process can only produce training examples. However, this definition seems restricted since when diffusion models generalize, they not only differ from the PF-ODE dynamics, but generate "high quality" samples as well. In other words, the V-kernel should have certain benign properties. This point is not adequately addressed in the current paper.
>
> This is a great point, and we strongly agree that it is important to address. We will address this point in two ways. First, we will add a **new section** to the main text ("Generalization through variance: consequences and examples") that explicitly discusses various "benign" properties of the V-kernel that support reasonable generalization. These properties include the fact that it tends not to add noise far from training data, that it tends not to increase the effective dimensionality of the training distribution (e.g., if training data lie in a plane, to good approximation so will the generalized distribution), and that training data is more likely to be sampled than novel states. The new section will also contain various numerical experiments that validate our results in simple 1D and 2D settings.
>
> Second, we will add a new appendix ("Benign properties of generalization through variance") that contains additional discussion of this point, including more math.
>
> > (ii) In section 5 the authors derive the analytical form of the V-kernels under three circumstances. However, the resulting equations (9), (13) and (17) are really hard for me to interpret. Can the authors provide more explanation on the unique properties of these different V-kernels and how they shape the distributions corresponding to the reverse sampling process?
>
> We agree that the results of our original draft are fairly formal, and that it would be extremely helpful to illustrate them using a series of concrete examples. The new main text section we are writing shows precisely what the V-kernel looks like for a variety of simple models trained on 1D and 2D point clouds, and in particular shows how generalization through variance is affected by (i) training set structure (specifically, gaps between training data and duplications of training data), (ii) anisotropy in the forward process, and (iii) the feature set used (in the case of linear models). Throughout this section, we will provide additional commentary linking the experimental results to our math results.
>
> > (iii) As a researcher focuses on empirical works, I don't find the results of this paper very useful. I think the paper could benefit from a further discussion on how the results can help improving practical diffusion models.
>
> Although our focus is more 'scientific' (i.e., understanding an observed phenomenon) than 'practical' (i.e., improving model performance), we plan to add comments related to improving practical models to the Discussion, and hope that the new experiments we add to the main text are also helpful. One finding of the experiments is that feature-related inductive biases appear to matter a lot in practice, with the choice of feature set, and how it interacts with the proxy score covariance, sometimes strongly affecting the form of generalization through variance.

---

### Official Review · Reviewer_hd21 · 2024-11-04

**Soundness:** 3
**Presentation:** 3
**Contribution:** 2
**Rating:** 6
**Confidence:** 3

**Summary:**

This paper investigates the generalization capabilities of diffusion models through the perspective of "generalization through variance," a novel concept where the inherent noise in the denoising score matching (DSM) objective used in training significantly influences model generalization. The authors develop a mathematical theory using a physics-inspired path integral approach, which reveals how the covariance structure of the noisy training target impacts inductive biases. The findings suggest that diffusion models are not merely reproducing training data but are capable of filling in 'gaps' that enhance their generative abilities.

**Strengths:**

The paper's originality lies in its novel theoretical framework, which is a significant leap in understanding the mechanics behind diffusion models' ability to generalize. The quality of the theoretical analysis is high, and the writing is clear. The significance of the work is evident as it gives a new understanding of the generalization of diffusion models.

**Weaknesses:**

1. The paper's main weakness is the lack of empirical validation of the theoretical claims. While the theoretical intuition is clear, the paper would be greatly strengthened by including simulations or experiments that demonstrate the practical impact of the "generalization through variance" phenomenon.
2. There is a lack of theory directly bridging the gap between variance and generalization error.

**Questions:**

**1.** On the bottom of page 1 “At present, there is arguably no theory that describes…” It seems there exists a bunch of papers analyzing the generalization ability of diffusion models including their generalization time and its relation to data structure, see [1], [2].

[1] Li, P., Li, Z., Zhang, H., & Bian, J. (2023). On the generalization properties of diffusion models. Advances in Neural Information Processing Systems, 36, 2097-2127.

[2] Fu, S., Zhang, S., Wang, Y., Tian, X., & Tao, D. (2024). Towards Theoretical Understandings of Self-Consuming Generative Models. arXiv preprint arXiv:2402.11778.

**2.** It would be clearer to add more description to Figure 1, including the scale of the variance heat map, etc.

**3**. How does the "generalization through variance" compare with other known mechanisms of generalization in diffusion models, such as those related to model capacity and training set structure? In particular, is there a specific formula between V-kernel with any kind of generalization error (or its bound)? If so, the generalization could be more quantitively measured based on V-kernel.

**4**. Could the authors provide empirical evidence or simulations that specifically illustrate the effects of the covariance structure on generalization as hypothesized in the paper?

---

> ### Author Response · Authors · 2024-11-28
> **Review response part 1**
>
> We thank the reviewer for their time and helpful comments. Before we address them, we would like to note that there is something of a qualitative distinction between two questions regarding how diffusion models generalize:
>
> 1. Given a finite number of samples M from some data distribution, what do diffusion models learn? (Relatedly: do models learn to simply regurgitate one of the $M$ samples? Do they learn something simple, like a kernel density estimate, or not?)
> 2. In the limit as $M \to \infty$, do diffusion models learn the ground truth distribution? (Relatedly: if yes, how does convergence depend on $M$?)
>
> These questions are both interesting and worthy of study. In our work, we focus almost exclusively on the first question. Our comment  “At present, there is arguably no theory that describes…” is intended to refer to the fact that, while a large number of papers address question 2 (including the Li et al. and Fu et al. papers you provide as examples), to our knowledge there is a theoretical gap regarding question 1.
>
> The distinction between the two may be somewhat confusing, as there are two asymptotic limits one can consider. One is the limit in which the model has access to an infinite number of samples from the data distribution (e.g., an infinite number of pictures of real dogs). The other is the limit in which, during training, the model has access to an infinite number $P$ of *training examples*, which consist of noise-corrupted versions of data distribution samples. We take the $P \to \infty$ limit but *not* the $M \to \infty$ limit, whereas most authors take both to infinity. This drastically changes the nature of the corresponding mathematical problem, and we think that this emphasis contributes to the novelty of our work.
>
> See below for a point-by-point response.
>
> **Weaknesses**
>
> > 1. The paper's main weakness is the lack of empirical validation of the theoretical claims. While the theoretical intuition is clear, the paper would be greatly strengthened by including simulations or experiments that demonstrate the practical impact of the "generalization through variance" phenomenon.
>
> We agree that this is a major weakness, and are working on a **new main text section** ("Generalization through variance: consequences and examples") before the Discussion to remedy this. The new section considers the impact of generalization through variance (GTV) in the context of a variety of 1D and 2D examples. While these examples are admittedly toy given that real diffusion models typically involve high-dimensional data, these simple examples are easy to visualize, they make the effects of GTV easy to see and understand, and they allow many quantities of interest to be straightforwardly computed. (They are also simple enough that we can consider a large number of models, since training is not an issue.)
>
> We show that generalization, as we claim, generally happens in regions where proxy score covariance is high, which corresponds to regions between multiple training examples. The details of this are modulated somewhat by the choice of feature maps in the linear case: for example, we show that rotating the data distribution and/or feature maps can yield a slightly different kind of generalization.

---

> > ### Author Response · Authors · 2024-11-28
> > **Review response part 2**
> >
> > > 2. There is a lack of theory directly bridging the gap between variance and generalization error.
> >
> > As we noted above, generalization error (which is typically viewed as a function of the data distribution size $M$) is not the main concern of our paper. However, we agree that the paper could be improved by some work in this direction. We have thought hard about this point, and are writing a **new appendix** section to address it.
> >
> > To us, the interesting generalization-error-related question is: how does the form of the V-kernel affect it? Since our main mathematical results merely specify the form of the effective reverse process, but not its solution (and hence, the precise form of $[q]$), this is far from obvious. Moreover, it is highly mathematically nontrivial to get results in this direction, since it is not possible to analytically solve most SDEs of interest (especially when the noise term is state-dependent, as here).
> >
> > We try to address this issue in two ways. First, we consider two special limits of the effective SDE: one in which most of the noise is added at the very end (the 'kernel density estimate' limit), and one in which the noise is small enough that that path integral is dominated by a single path (the 'semiclassical' limit, in physics terminology). In both cases, one can derive expressions for generalization error, although in the latter case it is still somewhat implicit.
> >
> > Second, we consider a tractable special case: we assume the ground truth is known to be Gaussian, and consider the NTK-associated V-kernel. In this case, $[q]$ and the generalization error can be computed exactly, and it can be shown that having a nontrivial V-kernel improves the speed of convergence somewhat.
> >
> > **Questions**
> >
> > > **1.** On the bottom of page 1 “At present, there is arguably no theory that describes…” It seems there exists a bunch of papers analyzing the generalization ability of diffusion models including their generalization time and its relation to data structure, see [1], [2].
> >
> > See our above comments: to our knowledge, there is a theoretical gap regarding generalization when the number $M$ of data distribution samples does not go to infinity. We consider a variety of examples with $M < 10$, which are almost certainly not addressed by asymptotic results. However, we agree that the cited papers (and other related work) are relevant, and that it is worth commenting more explicitly on how our results relate to theirs, so we will do this in a revised version.
> >
> > > **2.** It would be clearer to add more description to Figure 1, including the scale of the variance heat map, etc.
> >
> > Good point, we will make a few changes to improve clarity.
> >
> > > **3**. How does the "generalization through variance" compare with other known mechanisms of generalization in diffusion models, such as those related to model capacity and training set structure? In particular, is there a specific formula between V-kernel with any kind of generalization error (or its bound)? If so, the generalization could be more quantitively measured based on V-kernel.
> >
> > See our above response to weakness 2: we can come up with a formula that relates the two in a certain special case (the 'semiclassical' approximation, which assumes relatively small noise throughout), but the formula is still somewhat implicit.
> >
> > Conceptually, we do not view generalization through variance as separate from generalization related to model capacity and training set structure, but entangled together with them. One could only disentangle them in principle (we claim) by using a variant of the objective that does not involve randomness. Our NTK result (Prop. 5.2) explicitly includes a prefactor that relates to model capacity, and the training set structure strongly influences the form of the proxy score covariance, which is a key factor in all of our results.
> >
> > > **4**. Could the authors provide empirical evidence or simulations that specifically illustrate the effects of the covariance structure on generalization as hypothesized in the paper?
> >
> > We agree that this is crucial, and are currently working on various experiments. See our above response to weakness 1.

---

> > > ### Comment · Reviewer_hd21 · 2024-11-30
> > > **Response**
> > >
> > > Dear authors,
> > >
> > > Thank you for your response. I’ve updated my score accordingly.

---

### Official Review · Reviewer_DoAH · 2024-11-07

**Soundness:** 3
**Presentation:** 4
**Contribution:** 3
**Rating:** 6
**Confidence:** 3

**Summary:**

The main idea of the paper is that diffusion models generalize since the training step of estimating the score uses the “proxy score” as its target, which is a noisy version of the true score, or more accurately, a pointwise version of the score, conditioned also on the training data point. This introduction of this randomness into the nonlinear estimation process, and then into the diffusion process is claimed to lead to randomness in the generation process, and in turn, to generalization.

**Strengths:**

Understating the empirical success of diffusion models in creating new samples that are similar to the training set is a timely and central question. The analysis made in the paper is dedicated to the generative question, and does not impose generalization measures from other problems (e.g., the MSE of the score function). The intricate details of the diffusion models considered are clearly discussed, and how they affect the generalization process. The paper is well written and conveys its ideas in an interesting way.

An analytic stochastic path integral technique is used to derive closed-form expressions for the sampling distribution of new samples, with focus on the covariance structure, which highlights the role of the variance of the score proxy in the generation of new samples. It is also a direct performance measure compared to the MSE of the score. This technique is exemplified on two analyzable architectures.

**Weaknesses:**

1) The paper assures that the generated samples are different from the training samples due to the noise in the score proxy, but this does not seem to explain why the generated samples are meaningful in some sense (e.g., have similar features to the training samples). If one assumes a ground truth distribution then this is obvious, but the authors emphasize that they refrain from that, but it is not clear what replaces this assumption (beyond the generated sampled being different from the training samples).

2) Except for the appearance of a non zero covariance matrix in noise of the reverse diffusion process, the expressions in Propositions 5.1 and 5.2 are somewhat implicit. For example, it is not obvious how easy it is to compute them, or how its form affects generalization.

**Questions:**

1. Line 67: What is exactly meant by “boundary regions” ? One example is regions that are close to multiple training points, but how these are defined in general ? As high likelihood regions ? This term is used repeatedly afterwards so it would be good to accurately define it.

2. Line 75: How does the claim “models generalize better when they are somewhat underparameterized” agree with the (Karras et al, 2024) paper mentioned on line 48 ?

3. Line 78: Why interpolation considered a non-trivial generalization ?

4. Line 138: It would be better to introduce the proxy score before (4).

5. How is the operator \mathcal{D} is defined on (6) ?

6. Section 4: It is explained that even using the generated samples to estimate the score leads generalization. But what is the starting point of the process X_T (the noise point). Isn't this point generated from the training data in a noisy manner, and this also contributes to generalization ?

7. Line 207: Once the higher order terms in (7) are neglected, it is mentioned that this implies that the estimator distribution is approximately Gaussian. However, due to the Gaussianity of the forward process, doesn't this tacitly also approximate the data distribution as Gaussian ? In other words, if we assume that the data distribution is Gaussian, would the result be the approximated integral of (7) ?

---

> ### Author Response · Authors · 2024-11-28
> **Review response part 1**
>
> We thank the reviewer for their time and helpful comments. See below for a point-by-point response.
>
> **Weaknesses**
> > 1. The paper assures that the generated samples are different from the training samples due to the noise in the score proxy, but this does not seem to explain why the generated samples are meaningful in some sense (e.g., have similar features to the training samples). If one assumes a ground truth distribution then this is obvious, but the authors emphasize that they refrain from that, but it is not clear what replaces this assumption (beyond the generated sampled being different from the training samples).
>
> This is a good point: how do we know if a given generalization of training data is 'reasonable' without a ground truth? In the revised version of the paper, we address this issue in three ways.
>
> First, we describe various generic properties of the V-kernel that make it in some sense 'benign' (to borrow a phrase used by reviewer AyYq). These properties include the fact that it tends not to add noise far from training data, that it tends not to increase the effective dimensionality of the training distribution (e.g., if training data lie in a plane, to good approximation so will the generalized distribution), and that training data is more likely to be sampled than novel states. These properties are described both at the beginning of a **new main text section** ("Generalization through variance: consequences and examples") and a **new appendix** ("Benign properties of generalization through variance").
>
> Second, in the aforementioned new main text section, we depict a number of examples of generalization through variance (GTV) in 1D and 2D settings. Although these settings are admittedly toy, they are useful because they are easy to visualize, the results are easy to interpret, and it is straightforward to numerically compute all quantities of interest. These examples provide empirical evidence that the generalization achieved is in some sense reasonable. We underline the point that there are multiple ways to reasonably generalize a point cloud, though, by showing different types of generalization of the same 2D point cloud depending on the model and hyperparameters used.
>
> Third, we include a **new appendix** section (and limited discussion in the main text) on generalization error, as measured in terms of the KL-divergence between the ground truth distribution and the learned distribution, in settings where there is a known ground truth. It is hard to analyze except in special cases, but an interesting empirical insight is that error tends to fall faster as a function of the number of ground truth samples for diffusion models than comparable Gaussian-based kernel density estimates.
>
> > 2. Except for the appearance of a non zero covariance matrix in noise of the reverse diffusion process, the expressions in Propositions 5.1 and 5.2 are somewhat implicit. For example, it is not obvious how easy it is to compute them, or how its form affects generalization.
>
> This is a good point. We add additional commentary on computing them in practice as comments in the main text and relevant appendices. The V-kernel can be computed directly, since one only needs access to (i) the covariance of the proxy score (which we compute in Appendix B), whose computation involves computing a certain covariance; and (ii) the feature maps $\phi(\mathbf{x}, t)$ in the linear case. In the new main text section, we directly compute it in a number of cases. More generally, since the V-kernel has a form that depends on certain expectations, it is expected that it is also reasonably straightforward to compute in higher dimensions.
>
>
> **Questions**
>
> > 1. Line 67: What is exactly meant by “boundary regions” ? One example is regions that are close to multiple training points, but how these are defined in general ? As high likelihood regions ? This term is used repeatedly afterwards so it would be good to accurately define it.
>
> More generally, we mean values of $\mathbf{x}$ and t for which $p(\mathbf{x}_0 | \mathbf{x}, t)$ is highly uncertain (above some threshold, say), which (via Bayes' rule) can be converted into a statement about the curvature/Hessian of the noise-corrupted likelihood (or log-likelihood) $p(\mathbf{x} | t)$. These are precisely regions between multiple training points in the discrete case, but this definition also makes sense for other types of training distributions.
>
> We are adding a **new appendix** section describing what we mean by 'boundary regions' since the idea is central to our results, and since clarity about our usage of it will probably help other readers.

---

> > ### Author Response · Authors · 2024-11-28
> > **Review response part 2**
> >
> > > 2. Line 75: How does the claim “models generalize better when they are somewhat underparameterized” agree with the (Karras et al, 2024) paper mentioned on line 48 ?
> >
> > We should probably be a bit more careful here. What we really mean is: in the top-performing models of Karras et al. (2024), and also those of Karras et al. (2022), the number of samples used during training is somewhat greater than the number of model parameters. For example, the best class-conditional ImageNet-512 model has ~ 200 million parameters and is trained on ~ 2000 million samples. In Karras et al. (2024), Fig. 12 shows that overfitting happens without dropout beyond a certain model size, but the authors do not conclusively show what happens when the number of parameters is comparable to or larger than the number of model parameters.
> >
> > We will change our wording slightly to make what we're claiming clearer and more accurate.
> >
> > > 3. Line 78: Why interpolation considered a non-trivial generalization ?
> >
> > We probably could have been more precise here. In this setting, we consider "trivial" generalization to coincide with $p(x, t)$ at some small $t$ (like the reverse process cutoff $\epsilon$), i.e., to be a Gaussian kernel density estimate. It is trivial firstly because there is no need to train a diffusion model to sample from it, and secondly because its generalization performance (e.g., in terms of FID score) is empirically underwhelming.
> >
> > Interpolation in the sense of adding probability mass between data points, but not elsewhere, is not what such kernel density estimators do, and hence we consider it nontrivial. We will modify the text to be more clear on this point, and apologize for the confusion.
> >
> > > 4. Line 138: It would be better to introduce the proxy score before (4).
> >
> > Good catch, we have adjusted how it is introduced to fix this.
> >
> > > 5. How is the operator \mathcal{D} is defined on (6) ?
> >
> > This is notation for the path integral measure, which itself is shorthand for the measure of a large number of integrals. The bracket notation is unconventional, and probably confusing, so we have modified it to look more measure-like and clarified this in the text.
> >
> > > 6. Section 4: It is explained that even using the generated samples to estimate the score leads generalization. But what is the starting point of the process X_T (the noise point). Isn't this point generated from the training data in a noisy manner, and this also contributes to generalization ?
> >
> > Having a noisy starting point $x_T$ (which here and elsewhere is sampled from a Gaussian with variance $\sigma_T^2$) does not contribute to generalization, since integrating the PF-ODE with this initial condition and the true score ought to yield the data distribution (at least, for a small enough step size and reverse process cutoff). One needs extra noise from somewhere else to get something different than the data distribution: in Sec. 4, this is noise that comes from using the naive score estimator instead of the true score.
> >
> > We think the main text could have been clearer on this point, and will modify it somewhat. We will also add comments about an instructive limiting case of the naive score estimator, which allows one to more explicitly see how the V-kernel influences generalization.
> >
> >
> > > 7. Line 207: Once the higher order terms in (7) are neglected, it is mentioned that this implies that the estimator distribution is approximately Gaussian. However, due to the Gaussianity of the forward process, doesn't this tacitly also approximate the data distribution as Gaussian ? In other words, if we assume that the data distribution is Gaussian, would the result be the approximated integral of (7) ?
> >
> > No, and this is an important and subtle point. If the estimator $\hat{s}(x, t)$ is approximately Gaussian (in the sense that the value of $\hat{s}$ at some specific inputs $x$ and $t$ varies in a Gaussian fashion across training runs), we obtain Eq. 7. But this does *not* imply the data distribution is approximated as Gaussian. The data distribution is described by an effective SDE. Conditional on an initial condition $x_T$, this SDE does not necessarily yield a Gaussian distribution for the final point $x_0$, even though each small-time transition probability is Gaussian, because the noise term is generically state-dependent and the drift term is generically not linear.
> >
> > One would not obtain a Gaussian approximation of the data distribution unless two things are true: the score function is linear in $x$, and the V-kernel is independent of $x$. Neither is true in general. More broadly, one should keep in mind that the family of distributions describable as the solution to some set of SDEs is extremely rich.
> >
> > We have added comments clarifying this point in both the main text and the relevant appendices, since it is crucial for understanding what kind of approximation we're making. Experiments in our new main text section will show that this assumption is fairly reasonable.

---

> > > ### Comment · Reviewer_DoAH · 2024-12-03
> > > **Updated score**
> > >
> > > Thank you for the detailed answers. I have further increased the score, and keeping my positive judgment.

---

### Author Response · Authors · 2024-11-28
**Global reviewer response**

We thank all of the reviewers for their comments, which we feel have substantially improved and clarified the paper. We are still working on implementing the associated changes, so we will not be able to provide a modified PDF by the deadline, but the major and minor changes are as follows.

**Major changes.**

**New main text section.** We are adding a section titled "**Generalization through variance: consequences and examples**" after the main theoretical results and before the discussion. The goal of this section is to both empirically validate our mathematical results, and to (in the context of various simple experiments) show how generalization through variance affects distributions in practice. To ease both interpretability and computation, we are conducting these experiments in simplified 1D and 2D settings.

The specific layout of the section is as follows. First, we discuss various "benign" properties of generalization through variance (e.g., variance is not added far from training examples, and does not change the dimensionality of the data manifold) that suggest that it produces a 'reasonable' type of generalization. This is majorly due to a suggestion by reviewer AyYq.

Next, we conduct small experiments to show how generalization through variance relates to filling (or exaggerating) gaps between training examples, the existence of outliers and duplicates in training data, asymmetry in the forward process, and different feature sets (in the case of the linear score estimator). As previously mentioned, we consider only 1D and 2D point cloud data distributions here.

Finally, we comment on the relationship between the form of the V-kernel and generalization error, as suggested by reviewer hd21. This turns out to be a difficult problem to address mathematically in general, so we do this in a few special cases.

**Minor changes.**

- **New appendix on benign properties of generalization through variance.** This appendix provides more mathematical detail regarding why one generically expects generalization through variance to be 'reasonable' given its relationship to the proxy score covariance. This is due to a suggestion by reviewer AyYq.
- **New appendix on boundary regions, and how they relate to the proxy score covariance.** Because we use the concept of 'boundary regions' throughout the paper, we think it would be helpful to clarify what we mean. We discuss what we mean mathematically and relate this notion to concepts like "score blindness" which have been previously discussed in the diffusion model literature. This is due to a suggestion by reviewer DoAH.
- **New appendix on generalization error.** This appendix computes generalization error (which we formulate in terms of the KL-divergence between the typical learned distribution $[q]$ and some ground truth distribution $p_*$) in a few special cases, and in particular shows how it depends on the form of the V-kernel. The special cases of interest essentially correspond to when the reverse process noise is very large, and when it is very small. This is due to a suggestion by reviewer hd21.
- **New appendix presenting main results in terms of alternative formulations.** While we chose to present our main results in terms of score-matching, one can also derive similar results for other formulations of diffusion modeling. We present our main results (the three propositions from Sec. 4-5) for two other formulations used in practice: the 'denoiser' formulation (used by, e.g., Karras et al. 2022), and the 'noise prediction' formulation (used by, e.g., the original Stable Diffusion paper). This is mostly just a quality of life change to make our results more accessible to other researchers.
- **Clarification of various technical points.** We add additional commentary regarding certain technical issues, e.g., the relationship between assuming the score estimator distribution is Gaussian and assuming the data distribution is Gaussian (reviewer DoAH), and what kind of generalization counts as 'nontrivial' in this setting (also reviewer DoAH).
- **Small quality-of-life improvements.** We fix various small issues pointed out by reviewers, e.g., the introduction of the proxy score (reviewer DoAH), and adding more description to the Figure 1 caption (reviewer hd21).
- **Additional citations.** We have cited additional related work, including the two papers suggested by reviewer hd21.

We thank all reviewers for their patience as we work on implementing these changes and additions.

---

### Meta-Review · Area_Chair_m2Ec · 2024-12-22

**Metareview:**

This paper studies how diffusion models generalize when they are trained using denoising score matching. The analysis in the paper is intriguing and focuses on how the score is targeted with a noisy version (the conditional score) with regression rather than directly impacts the generalization. They then develop theory to understand this kind of generalization through variance. The paper is clearly written. There were weaknesses written by the reviewers around empirical validation and some of the assumptions, but the concerns were minor enough to have all the reviewers accept this paper.

**Additional Comments On Reviewer Discussion:**

All the reviewers were positive. The authors replies changed the opinion of several reviewers as well.

---

### Decision · Program_Chairs · 2025-01-22

Accept (Poster)